# Assessment of the Results and Methodology of the Sustainable Development Index for Spanish Cities

Javier García López [1,2,3,*] , Raffaele Sisto [1,2,3] , Javier Benayas [4] , Álvaro de Juanes [2,5], Julio Lumbreras [6] and Carlos Mataix [3]

1 Department of Organizational Engineering, Business Administration and Statistics, Escuela Técnica Superior de Ingenieros Industriales, Universidad Politécnica de Madrid, 28006 Madrid, Spain; industriales@upm.es
2 Smart & City Solutions SL, Calle Blasco de Garay 61, 28015 Madrid, Spain; info@smartandcity.com
3 Centro de Innovación en Tecnología para el Desarrollo Humano, Universidad Politécnica de Madrid (itdUPM), 28040 Madrid, Spain; itd@upm.es
4 Department of Ecology, Autonomous University of Madrid, 28049 Madrid, Spain; javier.benayas@uam.es
5 School of Economics and Business Administration, University of Alicante, c. San Vicente del Raspeig s/n, Building 31, 03690 San Vicente del Raspeig, Spain
6 Harvard Kennedy School, 79 John F. Kennedy St., Cambridge, MA 02138, USA; julio.lumbreras@hks.harvard.edu
* Correspondence: javier.garcialope@alumnos.upm.es

**Abstract:** In 2017, the United Nations adopted a global Sustainable Development Goals (SDG) indicator framework, calling on member countries to collect complementary national and regional indicators. Cities are crucial to channelling efforts towards sustainability through the use of these indicators. They provide an integrated approach to the city situation monitoring sustainability. However, more research is needed to understand how to adapt the goals, targets and indicators to specific municipal contexts. In 2020, the Spanish Sustainable Development Solutions Network launched the 2nd edition of the Spanish Cities Index. A set of 106 indicators allows for monitoring the implementation of the SDGs at the local level for Spanish cities. The objective is to perform a statistical audit to evaluate the consistency of the indicators and the impact of modelling assumptions on the result. The methodology used is an adaptation of the Handbook on Constructing Composite Indicator prepared by the European Commission. The indicator system is well balanced and covers the essential areas of the Sustainable Development Goals. The Spanish ranking is robust enough among the alternatives evaluated. However, some improvements are possible in the selection of indicators, e.g., removing redundant indicators and regional data. Finally, it is recommended to weigh goals based on municipal responsibility to adjust the results to the Spanish municipal context.

**Keywords:** sustainable development goals; SDG; 2030 Agenda; evaluation; indicators

## 1. Introduction

Based on the experience of the Millennium Development Goals (MDGs) [1,2], in 2015, the UN adopted the 2030 Agenda for Sustainable Development and its 17 Sustainable Development Goals (SDGs). They aim to guide the achievement of sustainable development [3] and are rank highly on the agenda of most countries in the world. The SDGs comprise 17 goals that cover different aspects of sustainable development under a holistic approach. These objectives, in turn, are further specified in 169 goals.

The evaluation and monitoring of sustainability through indicators are considered effective ways to condense the complex system dynamics, starting from a manageable amount of information used to evaluate the progress against the declared results [4].

Solid metrics and indicators are a practical sustainability measurement tool to assess progress and ensure achievement [5,6]. In 2017, the UN adopted a global framework of 247 indicators to assess progress in meeting the SDGs [7]. In this framework, member countries are asked to compile complementary national and regional indicators. The

implementation and success of this universal agenda require all levels of administration, the academic environment, civil society and the private sector [8,9]. In addition, each country is left free to establish its implementation strategies. Governments must be able to tailor targets and their indicators to fit national contexts and priorities. In this way, countries show their benchmarks against which they can evaluate their performance and help measure their progress.

In addition, these metrics should serve as a management tool for all the parties involved to carry out the necessary transformations to achieve the targets of the SDGs in 2030. For example, one of the first steps that countries must take is to establish voluntary monitoring evaluations of the progress made in each of the 17 SDGs [9]. The UN High-Level Political Forum plays a central role in monitoring progress globally [10].

It is estimated that more than two-thirds of the world's population will reside in urban areas by 2050, adding another 2.5 billion people to the current 4 billion urban residents [11]. Meeting the basic needs of growing urban populations while ensuring the integrity of their ecosystems, addressing climate change, and promoting economic productivity and social inclusion are the main challenges facing the cities of our time. They are considered places of critical importance for understanding and solving sustainability problems [12,13]. They are the primary consumers of energy [14], the largest generators of waste [14], and they produce the majority of global greenhouse gas emissions [15].

Urban planning decisions will play a critical role in achieving the SDGs [16]. In this sense, UN-Habitat has also developed an action framework of indicators specifically to assess the sustainability of cities [17]. This document examines the extent to which UN indicators will help cities assess their efforts to achieve results towards their sustainability. However, it does not provide either a policy roadmap for action or a data or policy monitoring system [18].

The recognition of the role of municipalities and local governments in facilitating sustainable development has led to a specific goal dedicated to cities and communities [19]. However, there are urban issues among the other 16 goals [20–22], and many cities already have their own sustainability goals. In particular, SDG 11 relates to sustainable and resilient cities and human settlements; given rapid urbanization, cities are generally recognized as key actors to implement the entire SDG agenda [17,19] successfully. It has thus become increasingly important to monitor their performance [23]. As urban systems are complex, a common way to simplify monitoring is by using indicators and their metrics [24]. Although they have been using indicators for a long time, it is only in the last few decades that an attempt has been made to compile sustainability indicators into sets that reflect the many different aspects required to assess their performance [25]. In this sense, the SDG indicators offer the possibility of a more balanced and integrated approach for monitoring urban sustainability [26,27].

To help countries in the annual balance of SDG progress, the Sustainable Development Solutions Network (SDSN) has been conducting yearly evaluations since 2016 through indices and dashboards of the Sustainable Development Goals [28]. Its evaluation report for countries, the SDG Index, presents a composite index that analyzes the 17 SDGs of the 2030 Agenda with 85 indicators. In its latest edition in 2020, it has included the analysis of 193 countries [29]. Likewise, the SDSN promotes evaluation reports and dashboards through its national and regional chapters evaluating progress in achieving the 2030 Agenda by measuring a series of indicators. It is an unofficial monitoring tool whose objective is to complement official efforts to monitor the 2030 Agenda implementation.

Table 1 shows the evaluation reports promoted by the SDSN and their distribution of indicators at the national, regional and local levels. SDSN reports distribute the aggregated indicators across the 17 SDGs to help countries and cities assess their degree of achievement and level of progress directly with the 2030 global political agenda [3].

**Table 1.** Number of indicators per goals in the reports developed by SDSN. (N = National; C = City).

| SDSN Report | Spatial Scope | SDG 1 | SDG 2 | SDG 3 | SDG 4 | SDG 5 | SDG 6 | SDG 7 | SDG 8 | SDG 9 | SDG 10 | SDG 11 | SDG 12 | SDG 13 | SDG 14 | SDG 15 | SDG 16 | SDG 17 |
|---|---|---|---|---|---|---|---|---|---|---|---|---|---|---|---|---|---|---|
| [30] | N | 4 | 6 | 18 | 3 | 7 | 4 | 4 | 5 | 6 | 1 | 4 | 6 | 4 | 7 | 4 | 9 | 5 |
| [31] | N | 3 | 7 | 16 | 5 | 7 | 4 | 4 | 6 | 7 | 2 | 4 | 3 | 4 | 6 | 5 | 9 | 4 |
| [32] | N | 3 | 6 | 18 | 4 | 7 | 5 | 4 | 5 | 6 | 2 | 4 | 5 | 4 | 5 | 4 | 10 | 5 |
| [33] | N | 3 | 7 | 16 | 6 | 8 | 7 | 5 | 8 | 6 | 1 | 2 | 8 | 4 | 4 | 3 | 14 | 3 |
| [34] | N | 4 | 5 | 20 | 9 | 6 | 6 | 3 | 8 | 9 | 3 | 7 | 5 | 4 | 4 | 6 | 11 | 3 |
| [35] | N | 3 | 6 | 20 | 9 | 6 | 6 | 3 | 8 | 9 | 3 | 7 | 6 | 3 | 6 | 6 | 10 | 3 |
| [28] | N | 2 | 6 | 11 | 6 | 5 | 3 | 4 | 6 | 7 | 3 | 3 | 3 | 2 | 5 | 3 | 7 | 3 |
| [36] | N | 3 | 6 | 15 | 5 | 5 | 4 | 4 | 6 | 9 | 3 | 3 | 8 | 4 | 5 | 5 | 9 | 5 |
| [37] | N | 3 | 6 | 17 | 8 | 5 | 6 | 4 | 6 | 11 | 3 | 4 | 8 | 5 | 6 | 5 | 9 | 5 |
| [38] | N | 3 | 8 | 17 | 9 | 6 | 7 | 4 | 7 | 10 | 3 | 4 | 7 | 5 | 4 | 5 | 10 | 5 |
| [29] | N | 3 | 8 | 17 | 9 | 6 | 7 | 4 | 7 | 10 | 3 | 4 | 7 | 4 | 5 | 5 | 10 | 6 |
| [39] | N | 7 | 7 | 14 | 6 | 7 | 4 | 5 | 8 | 8 | 6 | 5 | 6 | 9 | 0 | 4 | 7 | 0 |
| [40] | C | 2 | 2 | 5 | 7 | 2 | 3 | 1 | 3 | 1 | 1 | 3 | 2 | 1 | 0 | 2 | 2 | 2 |
| [41] | C | 2 | 2 | 5 | 6 | 3 | 3 | 2 | 3 | 1 | 2 | 7 | 2 | 2 | 0 | 2 | 2 | 2 |
| [42] | C | 3 | 3 | 12 | 5 | 6 | 3 | 5 | 5 | 5 | 6 | 10 | 4 | 3 | 4 | 3 | 5 | 3 |
| [43] | C | 3 | 3 | 7 | 4 | 2 | 2 | 2 | 3 | 2 | 3 | 10 | 1 | 1 | 0 | 2 | 3 | 1 |
| [44] | C | 3 | 2 | 8 | 4 | 3 | 1 | 1 | 4 | 2 | 3 | 7 | 1 | 1 | 0 | 2 | 2 | 0 |
| [45] | C | 5 | 3 | 7 | 4 | 3 | 3 | 3 | 3 | 3 | 5 | 5 | 3 | 3 | 0 | 3 | 4 | 0 |
| [46] | C | 2 | 1 | 6 | 7 | 3 | 2 | 1 | 3 | 6 | 1 | 10 | 4 | 1 | 0 | 4 | 5 | 0 |
| [47] | C | 4 | 4 | 9 | 6 | 4 | 3 | 4 | 7 | 5 | 3 | 3 | 0 | 3 | 0 | 2 | 3 | 2 |
| [48] | C | 2 | 5 | 19 | 21 | 5 | 5 | 1 | 6 | 2 | 6 | 3 | 2 | 2 | 1 | 1 | 4 | 2 |
| [49] | C | 5 | 5 | 13 | 6 | 5 | 6 | 4 | 8 | 6 | 6 | 11 | 5 | 4 | 5 | 4 | 9 | 4 |
| Average | | 3.21 | 4.95 | 13.11 | 6.11 | 5.11 | 4.21 | 3.32 | 5.47 | 6.21 | 2.84 | 5.42 | 4.68 | 3.37 | 3.21 | 3.84 | 7.26 | 2.89 |

A significant imbalance in each goal's number is observed by analyzing the distribution of indicators in these international and national reports. On the one hand, SDG 3 and SDG 16 have the highest number of indicators, followed by SDG 4 and SDG 9. On the other hand, SDG 10 and SDG 17 have the least number of indicators. In addition, significant differences can be observed between the distribution of indicators by SDG for country-level reports versus city-level reports. There are fewer indicators for each SDG compared to country-level reports due to the difficulties in finding data [13,50].

In the particular case of Spain, the *Red Española para el Desarrollo Sostenible* (REDS-SDSN) presented in 2020 the second edition of the Spanish Cities Report (SCR) [49] where more than 100 cities are evaluated and which is the object of analysis of this article. The study includes all the Spanish cities with more than 80,000 inhabitants and the regional capitals, covering over 50% of the total population in Spain. For this purpose, all the indicators selected were identified considering the national context and data availability of the official statistical sources.

The SCR maintains alignment with the global SDG framework similar to the SDSN's methodology for the SDG index. In this way, as with the countries, it is intended to help local Spanish entities to diagnose and evaluate their progress in each of the 17 SDGs. It presents a selection of aggregated indicators in the 17 objectives to link them with the 2030 Agenda. It has followed a rigorous selection and validation process run by representatives of the academic environment. It has also had the support of local entities and the Spanish Federation of Municipalities and Provinces. This report has become the benchmark for monitoring the progress of the objectives in the cities in the Spanish context.

In its latest edition of 2020, 106 indicators have been selected starting from the previous edition indicator set and following the SDSN methodology [51]. The indicators have been selected based on relevance, statistical adequacy, timeliness, quality and percentage of coverage. In addition, there has been a validation by experts for each SDG and a final public consultation to validate and rule out their suitability. These are essential aspects that contributed to increasing the transparency of the SCR.

The researchers of this article are also coauthors of the SCR. To continue with the research process, they have considered it necessary to evaluate and analyze the results obtained in greater depth. The main objective is to assess the robustness of the results and methodology of the Sustainable Development Index for the Spanish Cities. This could identify improvements for future editions by studying the impact of different alternatives in the calculation methodology and selecting indicators. For this, the methodology used by Joint Research Center (JRC) of the European Commission's Competence Center for the audit of the SDG index in 2019 [52] has been taken as a reference. In addition, based on the results obtained, the following complementary objectives are pursued: (i) consolidate its system of city indicators, (ii) analyze the results of different alternatives and (iii) validate their reliability.

This article does not intend to question the conceptual relevance of the indicator system. The aim is to analytically and objectively identify its main features and the improvement options that could be implemented based on the results obtained in its database.

## 2. Materials and Methods

In 2019, the SDSN requested an audit of the 2019 SDG Index from the Joint Research Center (JRC) of the European Commission's Competence Center on Composite Indicators and Scoreboards (COIN) [52]. This statistical audit focuses on two main issues: the statistical coherence of the structure of indicators and the impact of crucial modelling assumptions on the SDG Index ranking. This analysis was carried out in three stages: (i) Descriptive statistics of the data and data analysis to detect missing values and potential outliers; (ii) Multilevel analysis testing the statistical coherence of the structure and correlations between indicators and each SDG Index; (iii) Analysis of the index robustness and testing of the impact of crucial modelling assumptions on the SDG Index ranking. The JRC report also supplemented the country rankings of the SDG index with confidence intervals to better understand their robustness to the calculation method.

This JRC analysis has been taken by the authors as a methodological reference to achieve the objectives of this research, but applied to the SCR. This is possible because both reports use the SDSN methodology [51]. However, the Monte Carlo experiment was not performed to investigate the impact of varying the assumptions. Instead, to evaluate the effect of the weighting assumption, the survey published in the SCR report has been used. Thus, the analysis presented in this article follows these steps:

1.  Description and analysis of the indicators
    The objective is to identify potentially problematic indicators that could bias the overall index results. The authors used the same JRC rule to analyze the distributions [52]. An indicator should be considered for mathematical treatment if it has an absolute skewness more significant than 2.0 and a kurtosis greater than 3.5. In those cases, further analysis of their data distribution would be developed [53]. The formula for skewness is referred to as the Fisher-Pearson coefficient of skewness:

$$G1 = \frac{\sqrt{N(N1-1)}}{N-2} \frac{\sum_{i=1}^{N} \left( \frac{(Yi-\overline{Y})^3}{N} \right)}{s^3}, \tag{1}$$

The authors use the following definition of kurtosis:

$$kurtosis = \frac{\sum_{i=1}^{N} \left( \frac{(Yi-\overline{Y})^4}{N} \right)}{s^4} - 3, \tag{2}$$

2.  Alternatives for calculating the index, assumptions
    These methodological alternatives are collected in the assumptions listed below. The analysis results are intended to demonstrate how the choice of indicators and the

methodology affect the position of cities in the ranking. The selection of indicators and their targets can be considered two central points for defining the SDGs' performance metric [54]. This proves the sensitivity of rankings by comparing an Initial set ($I_s$) versus an Alternative set ($A_s$) of the SCR indicators. The assumptions raised in this study are the following:

    i.    Aggregation: arithmetic mean and geometric mean

        The geometric mean is usually used to aggregate heterogeneous variables and when the focus of the analysis is on percentage changes rather than absolute changes. For example, this method is used in the Human Development Index [55]. Its three-dimensional aggregation method for the arithmetic mean was changed to the geometric mean in 2010. Compared with the geometric mean, the arithmetic average has the advantage of the simplicity of interpretation: an index score between 0 and 100 reflects the average initial placement of the country between worst and best on the average of the 17 goals [51]. The study proposes the calculation of geometric mean for the Alternative set ($A_s$).

    ii.    Weighting of the SDGs

        The method for aggregating and weighting different variables into a single index can profoundly impact the overall ranking [56]. In the Initial set ($I_s$), each indicator was weighted equally. As a result, the relative weight of each indicator in a goal was inversely proportional to the number of indicators considered under that goal [51]. Different weightings of individual SDGs can have important implications on a city's performance and relative ranking in the composite index [57]. For this reason, the authors propose to use for the Alternative set ($A_s$) as the expert weight approach at the goal level [51], taking advantage of the survey on municipal competencies carried out by sustainability experts and members of local Spanish entities included in the report.

    iii.    Reduction of the indicator set

        To evaluate the statistical consistency of SCR indicators, a cross-sectional analysis is employed. The correspondence between the SCR index and real-world phenomena needs to be analyzed because correlations do not necessarily represent the real influence of the individual indicators on the phenomenon being measured [58]. The correlation aims to quantify the strength of the link joining two different indicators or goals [59]. Non-parametric correlation methods are commonly applied to those pairs of variables whose distribution is unknown a priori. This is the case of Spearman's analysis [60]. In contrast to Pearson's, the most commonly used correlation coefficient, Spearman's does not assume normally distributed and same-scaled variables [61]. This is why Pearson's rank has been used in several disciplines and previous studies [62,63]. The authors propose using the cross-correlation analysis to preliminary address the extent to which the data supports the conceptual framework [51]. The 1% significance level is used to determine whether the correlation between two variables is statistically significant [58]. To optimize and reduce the number of indicators, the Alternative set ($A_s$) will not include the correlated ones.

   3.    Analysis of the impact of assumptions:

    iv.    Principal component analysis

        Principal Component Analysis (PCA) is commonly used to assign weights to individual variables correlated and measured by a common underlying factor. In addition, PCA reduces the effects of multicollinearity by using a subset of the principal components in the model [51]. To analyze the impact of the previous assumptions using the Alternative set, the authors propose to use principal component analysis (PCA) to summarize each goal and interactions in the SCR. Applying PCA allows mapping trends, synergies and trade-offs at

the level of goals for all SDGs while using all available information on each indicator [64].

v.      Analysis of the variation of positions

This analysis aims to evaluate the shifts in the positions between the Initial set ($I_s$) and the Alternative set ($A_s$). Cities shifts under three positions cannot be considered significant, whereas differences of 10 places can show a meaningful difference [52]. The variation in the rankings, considering the previous assumptions, allows us to identify which cities show a particular sensitivity to changes.

## 3. Results

### 3.1. Description and Analysis of the Indicators

The SCR identifies a total of 106 indicators based on the ones from the 2018 edition. Of these indicators 84% (89 out of 106) have data at the municipal level, 47% (49 out of 106) are new to this edition or present improvements in their level of detail, and 60% (64 out of 106) have just been updated. The quality and reliability of the data stand out because they all come from official National and European statistical repositories or research centers and non-governmental organizations of recognized prestige. Data provided individually by the entities evaluated have not been accepted in the SCR to guarantee comparability and reliability. A full list of indicators can be found in Table A1 Appendix A.

The distribution of indicators for each SDG is balanced compared to other similar reports, as shown in Table 1. SDG 3 and SDG 11 with 13 and 11 indicators, respectively, stand out for their significantly higher number of indicators, while SDG 7, SDG 13, SDG 15 and SDG 17 hit a minimum of four. In general, data coverage for the indicators included in the index is suitable for all SDGs and all cities observed. In the particular case of the cities of the Basque Country region, some of the economic data is not available. SDG 14 is a particular case since the cities without coastal areas have no indicators for this goal.

Regarding the dataset provided, no specific issues have been found. Data did not require imputation for the index calculation because the selection of indicators already excluded those not reaching at least 80% coverage. Complete data can be found in Appendix B.

The SDSN methodology identifies the sustainability thresholds for each indicator based on the explicit/implicit goals of the SDGs, scientific goals or the average performance of the best actors and the specific criterion of the goal expert. At the same time, to eliminate the effect of extreme values and facilitate the comparability of results, the report authors have limited the data to the lower 2.5 percentile as the minimum value for normalization. The details of the specific values used on the maximum/minimum values and the chosen thresholds are described in Annex I of the SCR [49]. Indicator values are normalized using the minimum/maximum method from the dataset of all cities for any given indicator. The normalized value is then transformed into a value ranging from 0 to 100, which is directly comparable with the rest of the indicators. In other words, the city with the highest value of the raw data obtains a score of 100, while the lowest value will have a score of 0. This normalization operation guarantees that all the variables are ascending and, therefore, the highest values indicate positive performance in achieving each goal. It also eliminates outliers at both ends of the distribution because those cities that exceed the average of the best or worst values are assigned the same score, as recommended by the OECD manual for constructing composite indicators [58]. This improves their understanding and facilitates the communication of results.

The methodology used by JRC [52] for the SDG index analyzes skewness and kurtosis to assess the data distribution's shape and identify potentially problematic indicators. The rule applied by the JRC is that an indicator is valid for treatment if it has an absolute skewness greater than 2.0 and a kurtosis more significant than 3.5.

Table A2 in Appendix B shows potentially problematic indicators: 6 indicators with abnormal distributions (2b, 2c, 9d, 11h, 11i and 17a) and 11 indicators with negative

skewness (7d, 8h, 10f, 11c, 11d, 12d, 13a, 14b, 16a, 16i and 17d). As well as JRC, the authors applied different techniques to improve the distributions, such as logarithmic transformations, and their scatter plots have been analyzed in detail, but no significant improvements were observed. Finally, it has been decided to keep them in the calculation set due to their alignment with the official UN indicators [65] and to guarantee a minimum number of indicators per goal.

However, 16.04% (17 out of 106) of the indicators come from data at the provincial or regional level (see Appendix B). These indicators could alter the results because they do not reflect the particular reality of each city but rather a regional average. In the cities analyzed, several cities are very different from each other. Some of them are in single-province regions or with high depopulation rates. Some other cities belong to large metropolitan areas highly populated. Only 9 of the 17 SDGs present indicators with regional data, and there are never more than two indicators per goal. Therefore, its impact on the SCR index is relevant but limited.

### 3.2. Alternatives for Calculating the Index, Assumptions

#### 3.2.1. Aggregation: Arithmetic vs. Geometric Average

In the SCR, according to the SDSN methodology, the arithmetic average has been used as a two-stage aggregation method, at the indicator level for each goal and the goal level for the general index. An alternative aggregation method is proposed based on the geometric instead of the arithmetic mean to limit compensation between very different values in various areas of sustainable development [51]. Table 2 shows the position shifts in the SCR index obtained by changing from arithmetic to geometric average across SDG scores.

**Table 2.** Position shifts in the SCR Index. Alternative aggregation.

| Aggregation Method Shifts in the Index | Number of Cities | Percentage of Cities |
|:---:|:---:|:---:|
| 0 | 25 | 24.27% |
| 1 | 27 | 26.21% |
| 2 | 22 | 21.36% |
| 3 | 7 | 6.80% |
| 4 | 11 | 10.68% |
| 5 | 5 | 4.85% |
| 6 | 1 | 0.97% |
| 7 | 1 | 0.97% |
| 8 | 2 | 1.94% |
| 9 | 0 | 0.00% |
| >10 | 2 | 1.94% |

The two methods yield results that are almost the same and thus a nearly identical ranking. The volatility between ranks is minimal. These differences are due to the geometric average, which, unlike the arithmetic mean, penalizes significantly poor scores on specific goals. The maximum shift in positions is 10 and only occurs in two cities in the Madrid metropolitan area (1.94% of the total). Most cities, 74 out of 103 (71.84% of the total), change from zero to two positions. The cities that are most affected by the change in the aggregation method have their location in the metropolitan areas of Madrid, Catalonia, Basque Country and Andalusia.

#### 3.2.2. Weighting of the SDGs

The SDSN reports are calculated without using any type of weighting because all targets and SDGs are equally crucial for the 2030 Agenda by definition. Only the number of indicators per SDG skews their representativeness. However, assigning the same weight to the indicators and targets does not necessarily guarantee an equal contribution of the indicators or targets to the index results [58,66]. For example, the 13 indicators from SDG 3 and the 11 indicators from SDG 11 have less weight in the overall aggregation than the 4

indicators from SDG 7, SDG13, SDG 15 and SDG 17 (see Table 1). In conclusion, the greater the number of indicators per SDG, the less relative weight than other SDG indicators with a lower number.

The SCR [49] publishes an assessment of municipal competencies carried out by sustainability experts and members of local Spanish entities. The authors propose these results to create alternative weighting coefficients for normalizing the assessment values. These vary from 1.5 for the best result to 0.5 for the worst (Table A3 in Appendix C) to properly analyze their impact.

Table 3 shows the shifts in the position of the cities in the SCR index using this alternative weighting method. Of the total cities, 37.86% only change a maximum of two positions. Most cities, 59 out of 103 (57.28% of the total), change a maximum of four positions. The most affected cities by the alternative weighting method are sparsely populated southern cities of the peninsula and do not belong to any metropolitan area. However, unlike the application of the alternative aggregation method, this method significantly affects all cities and alters the results of the SCR index.

**Table 3.** Position shifts in the SCR Index. Alternative weighting.

| Weighting Method Shifts in the Index | Number of Cities | Percentage of Cities |
| --- | --- | --- |
| 0 | 7 | 6.80% |
| 1 | 12 | 11.65% |
| 2 | 20 | 19.42% |
| 3 | 7 | 6.80% |
| 4 | 13 | 12.62% |
| 5 | 6 | 5.83% |
| 6 | 3 | 2.91% |
| 7 | 8 | 7.77% |
| 8 | 9 | 8.74% |
| 9 | 5 | 4.85% |
| >10 | 13 | 12.62% |

3.2.3. Reduction of the Set of Indicators

The methodology proposed by the JRC [52] and the SDSN methodological paper [51] performs a correlation analysis to evaluate the statistical coherence of the SCR, aiming to reduce the set of indicators initially proposed. Determining the relationship and degree of dependency between the quantitative variables in the report evaluates the extent to which the data supports the index's conceptual framework. The analysis of the correlations (both positive and negative) between indicators makes it possible to identify redundancies, avoid an overvaluation of the same event, and, finally, reduce the model's complexity. The authors have analyzed the correlations of the indicators (with their SDG and with the general index) and the correlations of the SDGs (with each other and with the general index).

Table A4 in Appendix C shows the correlations between indicators with their respective SDG general index. Ideally, each indicator should correlate positively with its SDG and with the overall index. A significance level of 1% has been taken to determine if the correlation between two variables is statistically significant.

Indicators 4e and 11i (in red) show negative correlations with their SDG, but their coefficients are very low and not significant. These results are similar or even better than those obtained by the JRC analysis for the SDG Index in 2019. Only two indicators (1e—poverty line, and 13d—covenant of mayors) present a Pearson correlation coefficient higher than 0.92. It makes sense because they are of particular relevance to the achievement of the SDG. Furthermore, 21 of the 106 indicators present correlation coefficients higher than 0.70 and an acceptable significance level. Values greater than 0.70 are desirable as they imply that the index captures at least 50% ($\approx 0.70 \times 0.70$) of the variation in the underlying goals and vice versa [52]. In total, eight of the SDGs present two or more indicators with correlation values greater than 0.70, only three SDGs present a single indicator, and six



of the SDGs do not show any indicator with a correlation value greater than 0.70. This finding suggests that the selection of the indicators has been adequate because there is a low redundancy in the results [66].

Regarding the correlation with the general index, on the one hand, 20 negative correlations have been identified. They all have a very low correlation coefficient (<0.5), and only indicators 8g and 10d present acceptable levels of significance (<0.01). On the other hand, 10 indicators are identified with a positive correlation with a Pearson coefficient (>0.5) and an acceptable level of significance (<0.01). Therefore, SDG 1, SDG 4 and SDG 17 present a higher number of indicators with a better positive correlation, which corresponds to the highest scores in the city index. On the contrary, SDG 3 and SDG 13 present indicators with negative correlations and the worst scores of the cities that top the index.

Table A5 in Appendix C presents the Pearson coefficients of the 17 SDGs regarding the correlations at the SDG level. All of them correlate positively with the overall index. In addition, SDG 1, SDG 7, SDG 16 and SDG 17 show high positive and significant correlations with the index. Cities well positioned on these SDGs rank equally well in the SCR index. Furthermore, most of them (12 out of 17) present an excellent significance (<0.01). On the contrary, SDG 3 and SDG 14, with very low correlation coefficients, are identified as the worst-ranked cities in the SCR overall index.

Regarding the correlations between the Goals, only three of them have been identified with a high Pearson correlation coefficient (>0.50) and an acceptable level of significance (<0.01): (SDG 1 vs. SDG 4, SDG 7 vs. SDG 16, SDG 12 vs. SDG 17). Moreover, SDG 3 presents several negative correlations with other SDGs, but none have an acceptable significance level (<0.01). SDG 14 shows negative correlations with SDG 1 and SDG 4 with a low coefficient but a high significance level. Similarly, SDG 17 shows negative correlations with SDG 2 and SDG 14 with a low correlation coefficient but a high significance level.

Pearson correlation coefficients greater than 0.70 and significant values under 0.01 have been identified regarding the correlations between the indicators themselves. It shows a very high significant correlation which may suggest redundancy. The main values of the Pearson correlation analysis are summarized in Table A6 in Appendix C. Only a negative correlation has been identified between indicators 15a and 11j. The rest of the significant correlations are positive. To obtain a reduced set of indicators, those highly correlated with each other have been further analyzed to remove them [52]. Finally, the indicators removed for a reduced set are: 1d, 1e, 3f, 4d, 5a, 6e, 8e, 10a, 10f, 11d and 16h.

### 3.3. Analysis of the Impact of Assumptions

Based on the precedent results, variations in the methodology for calculating the SCR index can be proposed to evaluate their impact within a range of improvement alternatives. The objective is to quantify the uncertainty based on the difference in the position of the cities considered in the SCR index in each result. Table 4 shows three particular assumptions that have been identified in this uncertainty analysis. They are alternatives for the construction of the SCR index and can be easily investigated.

**Table 4.** Conceptual assumptions framework for the SCR Index.

| Assumption | Alternatives for $I_s/A_s$ |
|---|---|
| 1. Aggregation method (SDG level) | Arithmetic average/Geometric average |
| 2. Weighting method (SDG level) | SDG equal valuation/SDG valuation by experts |
| 3. Indicator set | Complete set/Reduced set |

According to the SDSN methodology, the arithmetic average has been used as an aggregation method in two stages in the SCR report: at the indicator level for each goal and the goal level for the overall index. In Section 3.2.1, the change in the aggregation method by geometric instead of arithmetic mean has been analyzed. It concludes that it does not significantly impact, so it has been ruled out as a suitable alternative for this study.

Consequently, the improvement alternatives to be analyzed are using a weighting method for the SDGs and using a reduction method for the indicators. Their impact analysis is carried out by comparing the initial set ($I_s$) of the SCR and the new alternative set ($A_s$) resulting from applying these alternatives. The evaluation of their results is carried out with two approaches: principal components analysis and an analysis of variation of positions in the index of cities.

### 3.3.1. Principal Component Analysis

Principal Component Analysis (PCA) aims to assess the extent to which statistical approaches confirm the conceptual framework [67]. It explores the correlation of all indicators simultaneously, highlighting, if present, some common trends that describe a common concept among the indicators [68,69]. The objective is to transform a set of original variables into a new set of variables that are a linear combination of the original ones, called Principal Components. These components or factors are unrelated to each other and successively explain most of the total variance. Ideally, it is expected to have one principal component explaining at least 70–80% of the total variance to claim a single latent phenomenon behind the data. As shown in Table A7 in Appendix C, this is not the case for the SCR Index. The results identify that six principal components explain almost 70% of the variance.

Eighty-two indicators are available for each city in the sample and seventeen intermediate indices referring to each SDGs and the overall index. Based on the 17 variables, a reduction of dimensions is carried out through a PCA. Table A6 shows the PCA results for $I_s$ and $A_s$. The eigenvalues represent the amount of variance explained by each factor; therefore, the higher the eigenvalue, the more variance each factor explains. The Kaiser–Guttman rule [70] has been used in this study due to its strict scale. It suggests keeping those factors with eigenvalues greater than 1.0. It would hold a total of six factors that would represent 66.94% of the explained variance. On the other hand, in $A_s$, the number of factors becomes seven, with an explained variance of 71.08%. In addition, the total explained variance and the distribution between the same factors increase, being more uniform among the seven factors of $I_s$ concerning those of $A_s$.

Consequently, the modification of indicators from one set to another has increased the sample's representativeness, at least in those latent relationships within the dataset.

In addition, Table A8 in Appendix C shows the rotated component matrix for both sets. They differ considerably concerning the composition of their factors, and none of these components exhibit a clear or logical arrangement concerning the subject discussed. Figure A1 in Appendix C shows a heterogeneous disposition to the issue analyzed. For factor 1, a group of SDG 12, SDG 16, SDG 17, SDG 4 and SDG 7 was analyzed versus an opposing group consisting of SDG 14 and SDG 10 (as suggested by negative correlations). For factor 2, ODS 1 is diametrically opposite to ODS14 and ODS15.

Consequently, it is necessary to visualize the composition of the intermediate indices and analyze them individually as has been done as a whole. Table A9 in Appendix C shows the analysis of the main components of the indicators that comprise each SDG. Factors indicate the number of factors generated by Factor Analysis; indicators indicate the number of indicators collected in the database for each SDG; % Variance is the total percentage of variance explained by these factors. The last column shows the variance difference explained within the alternatives.

Accordingly, 14 of the 17 SDGs contain a total explained variance more significant than 60%, exhibiting high representativeness and assessing the subject to be observed. This is reduced in $A_s$ where the number of SDGs with a total explained variance above 60% is reduced to five. The elimination of indicators in several SDGs has a severely negative impact. However, the cases that remain above the established criterion, i.e., SDGs 2, 3, 4, 8 and 11 that maintain a total explained variance above 60%, manage to describe and monitor the central theme of each of the goals. Therefore, the analytical loss of reducing an average of two indicators for each group of indicators per SDG does not compensate

or enrich the analysis in some of them. As can be observed, up to 34% of the analytical information losses for some SDGs do not interact well with reducing indicators.

Looking individually at the rotated component matrices of each of the SDGs, it is possible to observe groupings that explain different aspects within each goal. SDGs 3, 4, 8 and 11 are worthy of analysis because they obtain high total explained variances, and they present exciting relationships within their matrix:

SDG 3 is a goal with a large number of indicators and factors. Table A10 in Appendix C shows the rotated component matrix, filtering out those relationships equal to or greater than 0.3 in absolute value. Values represented only in one factor, being absolute relationships with their associated factor, are highlighted in bold. It can be observed that these relationships coincide with the most robust relationships in the matrix (except one existing in indicator n_sdg03_alcohol) and have a positive relationship with the measurement of SDG 3.

Figure A2 in Appendix C represents a rotated space component graph for SDG 4. It visually shows the type of relationship that the indicators that make up the SDG maintain with the calculated factors. It can be seen that there are two types of indicators or aspects within the SDG itself. Thus, factor 1, which contains most of the variance, is highly related to education expenditure per capita, an explicit effort that transversally influences the SDG. On the other hand, those outcome indicators that would give us a picture of the situation in the territory are grouped in factor 2, suggesting that they are different dimensions to be considered but not contrary or exclusive.

The analysis of SDG 8 (Table A11 in Appendix C) shows a first factor that explains 24.41% of the variance and represents a positive dimension for the SDG. Their indicators that have high relationships connect with the economic progress and productivity of the territories. In contrast, factor 2, which explains 23.06% of the total variance, is the one that represents the negative weighting of the SDG. These are the variables that have an inverse influence on the goal's progress, all referring to the unemployment data.

Finally, Table A12 in Appendix C presents the analysis of SDG 11. It does not exhibit explicit specializations or differentiated aspects within each factor and groups the variables that measure air quality in the same factor, which provides logic to the composition of this factor but does not present a valuable interpretation within the analysis.

### 3.3.2. Index Position Shifts Analysis

Table 5 summarizes the changes in the position of the cities in the SCR index between $I_s$ and $A_s$. Appendix C includes the complete list of cities and the index score for each variation of the calculation.

**Table 5.** Position shifts in the SCR Index. All alternatives included.

| Shifts in the index $I_s$ to $A_s$ | Number of Cities | Percentage of Cities |
|:---:|:---:|:---:|
| 0 to 5 | 17 | 16.50% |
| 6 to 10 | 25 | 24.27% |
| 11 to 15 | 36 | 34.5% |
| 16 to 20 | 10 | 9.71% |
| >21 | 15 | 14.56% |

It can be observed that 34.95% of cities (36 out of 103) change between 11 to 15 positions in the SCR index. Additionally, 24.27% (25 out of 103) change from 6 to 10 positions, 16.50% (17 out of 103) change from 0 to 5, 9.71% (10 out of 103) change from 16 to 20 positions and 14.56% change more than 21 positions.

The first three cities of the initial index ($I_s$) (Vitoria-Gasteiz, Zaragoza and Logroño) and the alternative index ($A_s$) maintain their positions. Similarly, the first five and the last five positions remain stable and do not change more than three positions between the different calculation alternatives. Table A13 shows the top 10 positions of cities on

each calculation alternative. Table A14 shows the bottom 10 positions of cities on each calculation alternative. The full list of results is in Table A15 in Appendix C.

## 4. Final Conclusions and Discussions

The SCR, like the SDG Index, proposes a one-of-a-kind composite measure to track the progress of the SDGs at the city level. A deep understanding of their underlying components and the relationships between them must accompany the results. The effort of cities, strategic territories for their contribution to the national socioeconomic and environmental performance [25], is essential to achieve compliance with the SDGs since the municipal level is closest to the daily lives of people and companies. Therefore, the adaptation of its policies to the 2030 Agenda and the measurement of its progress is urgent and necessary for the country's progress towards meeting the SDGs.

The SCR ranking is robust enough among the alternatives evaluated based on the previous evaluation of the results and the methodology. The sensitivity analyses performed confirm that the uncertainty is manageable. For this reason, it can be concluded that the system of city indicators is consolidated. However, according to [71], many indicator initiatives are driven by the availability of relevant and reliable data [72–74]. The limitation in the data availability conditions the use of the appropriate indicators [75]. In the case of SCR, the sets of indicators are biased and incomplete to measure sustainability. This situation jeopardizes the reliability of the results. Therefore, developing further scientific research and expanding the data collection at the city level is necessary. It is also hopeful that, as the availability of data increases to measure some of the goals, implicit weighting would be reduced across goals.

Regarding selecting indicators, two aspects should be improved to reduce the complexity of the evaluation system. On the one hand, redundancy between collinear indicators should be avoided because it is equivalent to double-counting the same urban phenomenon. This target seems to have been accomplished for the SCR. However, the indicators selected for the SCR should be positively correlated with each of the objectives they represent. The results in this aspect are slightly better than those obtained by the JRC analysis for the global SDG Index. These suggest that there is little redundancy in the indicators' data, and their selection has been correct because they measure different aspects of the city. Therefore, its representativeness is adequate and, from a statistical point of view, with low levels of uncertainty.

On the other hand, whenever possible, regional data should be omitted [50]. By repeating the data of cities in the same province, the singularities of each city are neglected. There are single regions with high depopulation in the cities considered, and others belong to large, highly populated metropolitan areas. In this way, very different city realities are mixed due to the chosen population and representativeness bias. In addition, the results of very different realities are simplified, and their comparability is difficult. It would be advisable to carry out specific analyses by regions or similar urban areas to deepen and broaden the results. Furthermore, it would be desirable to use compliance thresholds based on other criteria for selecting and grouping cities to complement the SCR. For instance, in addition to the number of inhabitants and representativeness, economic biases or population density could be used to make groupings between equals and improve comparability [76,77].

Regarding the calculation methodology, it can be concluded that the use of an alternative aggregation method from the geometric mean instead of the arithmetic does not significantly affect the index positions [51]. However, using an alternative weighting method has been shown to affect index positions significantly. The first and last positions of the index are not affected by this change in the weighting method, but the rest of the intermediate positions are. There is no specific pattern of cities that are more sensitive to this change. Further investigation would be necessary considering other variables.

The SCR index is based on the 2030 Agenda for Sustainable Development adopted by all UN member states and rigorously follows the same structure of 17 goals. The

indicator system is well balanced and covers the essential areas of the SDGs. However, as it is a framework designed at the country level, its application in cities requires an adaptation process. Therefore, indicators must adjust to the competence frameworks distributed among the different administrative levels and eminently urban phenomena. According to [78], this recent research contributes to a strong grounding for the successfully implementation of the SDGs in Spain at both the national and city levels. Corroborating with previous research, our findings show that no SDG can individually make a country evolve and comply with the 2030 Agenda but working with the SDGs as a whole can create a virtuous cycle of SDG progress. Once the datasets and indicators are consolidated and improved, it would be advisable to investigate the synergies and trade-offs between the results at the country level and the results of their main cities.

**Author Contributions:** Conceptualization, J.G.L., R.S. and J.B.; methodology, J.G.L. and R.S.; validation, J.L. and C.M.; formal analysis, J.G.L., R.S. and Á.d.J.; investigation, J.G.L. and R.S.; resources, R.S., J.G.L. and J.B.; data curation, Á.d.J.; writing—original draft preparation, J.G.L. and R.S.; writing—review and editing, J.G.L., R.S., Á.d.J. and J.L.; visualization, J.G.L. and Á.d.J.; supervision, J.B., J.L. and C.M.; project administration, J.G.L. All authors have read and agreed to the published version of the manuscript.

**Funding:** This research received no external funding.

**Data Availability Statement:** Not applicable.

**Acknowledgments:** The authors would like to special thank the Spanish SDSN Network (REDS) for organizational and technical support, and Ana Justel (Autonomous University of Madrid) and Alberto Quintanilla (Smart & City Solutions) for programming assistance. We are grateful for the support provided by the project "Hacia la consolidación de ciudades inclusivas, un desafío para Madrid—H2019/HUM-5744)", and anonymous reviewers for their feedback to improve this paper.

**Conflicts of Interest:** The authors declare no conflict of interest.

## Appendix A

**Table A1.** Full list of indicators included in the 2020 SCR.

| Goal | Id | Indicator | Description | Scale |
|---|---|---|---|---|
| | | | SDG 1—No poverty | |
| 1 | 1a | sdg01_2020ratio | 20:20 ratio. Income inequality metric | Municipal |
| 1 | 1b | sdg01_gastosocial | Social Welfare spending per capita | Municipal |
| 1 | 1c | sdg01_pobreza | Population with income per consumption unit below 40% | Municipal |
| 1 | 1d | sdg01_pobrezamenores | Child poverty rate | Municipal |
| 1 | 1e | sdg01_riesgopobreza | Risk of poverty rate | Municipal |
| | | | SDG 2—Zero hunger | |
| 2 | 2a | sdg02_agricultura | Organic farming rate | Provincial |
| 2 | 2b | sdg02_consumo | Food consumption prices index | Provincial |
| 2 | 2c | sdg02_empagri | Employment rate in agriculture and fishery | Municipal |
| 2 | 2d | sdg02_expagrariasurbano | Agricultural and Forestry Operations | Municipal |
| 2 | 2e | sdg02_supcultivos | Share of land of agricultural areas | Municipal |
| | | | SDG 3—Good Health and well-being | |
| 3 | 3a | sdg03_adfertility | Adolescent fertility rate | Municipal |
| 3 | 3b | sdg03_alcohol | Alcohol and drugs death rate | Municipal |
| 3 | 3c | sdg03_gripe | Infectious disease of the respiratory system death rate | Municipal |
| 3 | 3d | sdg03_hepatitis | Viral hepatitis-related deaths rate | Municipal |
| 3 | 3e | sdg03_infantil | Infant mortality | Municipal |
| 3 | 3f | sdg03_ncd | Non-communicable diseases death rate | Municipal |
| 3 | 3g | sdg03_prematuras | Premature mortality (under 65 years) | Municipal |
| 3 | 3h | sdg03_suicidios | Suicide death rate | Municipal |
| 3 | 3i | sdg03_trafico | Road traffic death rate | Municipal |

**Table A1.** *Cont.*

| Goal | Id | Indicator | Description | Scale |
|------|-----|-----------|-------------|-------|
| 3 | 3j | sdg03_tuberculosis | Tuberculosis death rate | Municipal |
| 3 | 3k | sdg03_tumores | Respiratory system tumours death rate | Municipal |
| 3 | 3l | sdg03_vida | Life expectancy | Municipal |
| 3 | 3m | sdg03_vih | HIV and AIDS death rate | Municipal |
| SDG 4—Quality education | | | | |
| 4 | 4a | sdg04_estudiantes | Students enrolled in higher education | Municipal |
| 4 | 4b | sdg04_gastoedu | Education spending per capita | Municipal |
| 4 | 4c | sdg04_guarderia | Children 0–4 in day care or school | Municipal |
| 4 | 4d | sdg04_isced012 | Adults with primary education (ISCED level 0,1–2) | Municipal |
| 4 | 4e | sdg04_isced34 | Adults with secondary education (ISCED level 3–4) | Municipal |
| 4 | 4f | sdg04_isced56 | Adults with higher education (ISCED level 5–8) | Municipal |
| SDG 5—Gender equality | | | | |
| 5 | 5a | sdg05_brechapension | Gender subsidy gap | Provincial |
| 5 | 5b | sdg05_brechasalarial | Gender salary gap | Provincial |
| 5 | 5c | sdg05_delitossex | Violence and sexual exploitation rate | Municipal |
| 5 | 5d | sdg05_denuncias | Gender violence rate | Municipal |
| 5 | 5e | sdg05_paridad | Seats held by women in municipal governments | Municipal |
| SDG 6—Clean water and sanitation | | | | |
| 6 | 6a | sdg06_balanceagua | Balance in budgets for water service | Municipal |
| 6 | 6b | sdg06_canon | Fee for water supply and sanitation rate | Municipal |
| 6 | 6c | sdg06_esfuerzo | Financial exertion for water supply | Municipal |
| 6 | 6d | sdg06_litros | Volume of water distributed per day | Municipal |
| 6 | 6e | sdg06_precioabasteci | Water supply price | Provincial |
| 6 | 6f | sdg06_preciosaneamiento | Water sanitation price | Provincial |
| SDG 7—Affordable and clean energy | | | | |
| 7 | 7a | sdg07_eficiencia | Reduction in spending on street lighting since 2012 | Municipal |
| 7 | 7b | sdg07_facturaelectr | Impact of electricity costs on average household income | Municipal |
| 7 | 7c | sdg07_renovable | Renewable energy rate | Provincial |
| 7 | 7d | sdg07_suministro | Power supply quality index | Provincial |
| SDG 8—Decent work and economic growth | | | | |
| 8 | 8a | sdg08_accidentes | Accidents at Work | Provincial |
| 8 | 8b | sdg08_desempleo | Unemployment rate | Municipal |
| 8 | 8c | sdg08_desempleocovid | Impact of COVID-19 on unemployment rate | Municipal |
| 8 | 8d | sdg08_desempleojovenes | Youth unemployment rate | Municipal |
| 8 | 8e | sdg08_desempleolarga | Long term unemployed | Provincial |
| 8 | 8f | sdg08_diversidad | Sector-dependency job index | Municipal |
| 8 | 8g | sdg08_pibcapitamun | GDP annual growth rate | Municipal |
| 8 | 8h | sdg08_productividad | Annual productivity growth rate | Municipal |
| SDG 9—Industry, Innovation and Infrastructure | | | | |
| 9 | 9a | sdg09_3g4g | 3G and 4G networks access index | Provincial |
| 9 | 9b | sdg09_bandaancha | Broadband penetration rate | Provincial |
| 9 | 9c | sdg09_empindus | Employees in Industry rate | Municipal |
| 9 | 9d | sdg09_gastoidi | R&D spending per capita | Municipal |
| 9 | 9e | sdg09_patentes | Patent applications local rate | Municipal |
| 9 | 9f | sdg09_sueloactecon | Land area planned for economic activities | Municipal |
| SDG 10—Reduced inequality | | | | |
| 10 | 10a | sdg10_mediana | Population under poverty line | Municipal |
| 10 | 10b | sdg10_discapacitados | People with disabilities in labor market | Provincial |
| 10 | 10c | sdg10_extranjeros | Foreign employment rate | Provincial |
| 10 | 10d | sdg10_igini | Gini index | Municipal |
| 10 | 10e | sdg10_indicedependencia | Child and elderly dependency ratio | Municipal |
| 10 | 10f | sdg10_top1 | Top 1%. Income inequality metric | Municipal |

**Table A1.** *Cont.*

| Goal | Id | Indicator | Description | Scale |
|------|-----|-----------|-------------|-------|
| | | | SDG 11—Sustainable cities and communities | |
| 11 | 11a | sdg11_no2 | NO$_2$ concentration. Air Quality indicator | Municipal |
| 11 | 11b | sdg11_o3 | Ozone concentration. Air Quality indicator | Municipal |
| 11 | 11c | sdg11_pm10 | PM10 concentration. Air Quality indicator | Municipal |
| 11 | 11d | sdg11_pm10dias | Days that exceed PM10 limits | Municipal |
| 11 | 11e | sdg11_pm10media | PM10 annual average | Municipal |
| 11 | 11f | sdg11_preciovivienda | Housing access index | Municipal |
| 11 | 11g | sdg11_residencias | Nursing home places | Provincial |
| 11 | 11h | sdg11_resiliencia | Urban resilience index | Municipal |
| 11 | 11i | sdg11_suptrans | Access to public transport index | Municipal |
| 11 | 11j | sdg11_viviendaprotegida | Access to protected housing | Provincial |
| 11 | 11k | sdg11_vulnerables | Urban vulnerability index | Municipal |
| | | | SDG 12—Responsible consumption and production | |
| 12 | 12a | sdg12_envases | Plastic recycling and packaging rate | Municipal |
| 12 | 12b | sdg12_impropios | Improper waste rate | Municipal |
| 12 | 12c | sdg12_papel | Paper recycling rate | Municipal |
| 12 | 12d | sdg12_turismo | Sustainable tourism | Municipal |
| 12 | 12e | sdg12_vidrio | Glass recycling rate | Municipal |
| | | | SDG 13—Climate action | |
| 13 | 13a | sdg13_CO2buildings | Buildings and industry CO$_2$ emissions per capita | Municipal |
| 13 | 13b | sdg13_CO2capita | CO$_2$ emissions per capita | Municipal |
| 13 | 13c | sdg13_CO2transport | Transportation CO$_2$ emissions per capita | Municipal |
| 13 | 13d | sdg13_medicion | Covenant of mayors for climate and energy network | Municipal |
| | | | SDG 14—Life below water | |
| 14 | 14a | sdg14_banderaazul | Blue flags index for coastal areas | Municipal |
| 14 | 14b | sdg14_calidad | Bathing sites with excellent water quality | Municipal |
| 14 | 14c | sdg14_costamun | Land built on the coastal strip of the first 500 m | Municipal |
| 14 | 14d | sdg14_dpmt | Protected public land–maritime domain | Municipal |
| 14 | 14e | sdg14_habitatsmun | Coastal and marine protected natural habitats | Municipal |
| | | | SDG 15—Life on land | |
| 15 | 15a | sdg15_cobartificial | Territory and habitat diversity. Artificial cover | Municipal |
| 15 | 15b | sdg15_enp | Protection of Natural Areas | Municipal |
| 15 | 15c | sdg15_zonaforestal | Forest areas | Municipal |
| 15 | 15d | sdg15_zonasverdes | Tree Cover Density | Municipal |
| | | | SDG 16—Peace, justice and strong institutions | |
| 16 | 16a | sdg16_blanqueo | Drug traffic crime rate | Municipal |
| 16 | 16b | sdg16_criminalidad | Crime rate | Municipal |
| 16 | 16c | sdg16_homicidios | Murders and violent deaths | Municipal |
| 16 | 16d | sdg16_participa | Voter turnout in municipal elections | Municipal |
| 16 | 16e | sdg16_participacion | Citizen participation and collaboration index | Municipal |
| 16 | 16f | sdg16_solidez | Strength and autonomy of the municipal institution | Municipal |
| 16 | 16g | sdg16_transparencia | Municipal transparency index | Municipal |
| 16 | 16h | sdg16_transparenciaeco | Economic and financial transparency index | Municipal |
| 16 | 16i | sdg16_violencia | Violence against children (under 13 years) | Provincial |
| | | | SDG 17—Partnership for the goals | |
| 17 | 17a | sdg17_coop | Cooperation and development projects | Municipal |
| 17 | 17b | sdg17_opendata | Open data index | Municipal |
| 17 | 17c | sdg17_redes | National networks to achieve the SDGs | Municipal |
| 17 | 17d | sdg17_zonasblancas | White NGA areas | Municipal |

# Appendix B

**Table A2.** Complete statistics of the indicators for the 2020 SCR.

| Goal | Indicator | Number of Cities | Missing Data (%) | Mean | Skewness | Kurtosis | Deviation | Variance |
|------|-----------|------------------|------------------|------|----------|----------|-----------|----------|
| 1 | 1a | 98 | 4.85 | 13.32 | 1.44 | 1.27 | 18.1 | 327.55 |
| 1 | 1b | 101 | 1.94 | 22.8 | 1.6 | 3.23 | 19.91 | 396.25 |
| 1 | 1c | 98 | 4.85 | 50.72 | −1.04 | 0.56 | 18.44 | 340.08 |
| 1 | 1d | 98 | 4.85 | 40.38 | −0.24 | 0.27 | 16.9 | 285.75 |
| 1 | 1e | 98 | 4,85 | 43.54 | −0.64 | −0.28 | 18.36 | 336.94 |
| 2 | 2a | 103 | 0 | 54.01 | −0.2 | −0.54 | 25.21 | 635.29 |
| 2 | 2b | 94 | 8.74 | 13.1 | 2.84 | 8.52 | 20.41 | 416.69 |
| 2 | 2c | 103 | 0 | 11.58 | 2.94 | 9.32 | 19.62 | 384.77 |
| 2 | 2d | 103 | 0 | 37.71 | 0.44 | −0.85 | 29.36 | 861.83 |
| 3 | 3a | 103 | 0 | 56.08 | 0.1 | 0.05 | 22.92 | 525.29 |
| 3 | 3b | 103 | 0 | 79.61 | −1.23 | 1.04 | 24.91 | 620.35 |
| 3 | 3c | 103 | 0 | 58.62 | −0.4 | −0.34 | 24.76 | 612.91 |
| 3 | 3d | 103 | 0 | 71.14 | −1.08 | 0.78 | 25.09 | 629.3 |
| 3 | 3e | 94 | 8.74 | 67.42 | −0.94 | 0.54 | 21.85 | 477.43 |
| 3 | 3f | 103 | 0 | 60.5 | −0.28 | −0.48 | 24.25 | 588.1 |
| 3 | 3g | 103 | 0 | 56.98 | −0.19 | −0.8 | 26.08 | 680.26 |
| 3 | 3h | 103 | 0 | 50.7 | 0.17 | −0.55 | 22.97 | 527.68 |
| 3 | 3i | 102 | 0.97 | 98.71 | −1.57 | 2.49 | 1.16 | 1.34 |
| 3 | 3j | 103 | 0 | 76.63 | −1.22 | 0.95 | 26.42 | 698.21 |
| 3 | 3k | 103 | 0 | 57.3 | −0.23 | −0.66 | 25.97 | 674.5 |
| 3 | 3l | 103 | 0 | 90.34 | −0.16 | −0.4 | 4.58 | 20.98 |
| 3 | 3m | 103 | 0 | 67.04 | −0.52 | −0.7 | 29.32 | 859.45 |
| 4 | 4a | 92 | 10.68 | 27.24 | 1.2 | 1.87 | 22.78 | 518.9 |
| 4 | 4b | 101 | 1.94 | 41.91 | 0.71 | −0.62 | 30.33 | 919.66 |
| 4 | 4c | 94 | 8.74 | 29.63 | 0.68 | 1.96 | 14.39 | 207.02 |
| 4 | 4d | 89 | 13.59 | 35.84 | 0.34 | 0.55 | 17.09 | 292.18 |
| 4 | 4e | 94 | 8.74 | 85.23 | −0.27 | −0.42 | 10.41 | 108.41 |
| 4 | 4f | 94 | 8.74 | 49.04 | 0.3 | 0.2 | 19.39 | 375.97 |
| 5 | 5a | 98 | 4.85 | 26.77 | −0.1 | 0.14 | 10.19 | 103.84 |
| 5 | 5b | 98 | 4.85 | 24.23 | 0.5 | −0.44 | 15.43 | 237.98 |
| 5 | 5c | 103 | 0 | 45.36 | −0.48 | −0.19 | 18.18 | 330.47 |
| 5 | 5d | 97 | 5.83 | 54.59 | −0.9 | 0.23 | 21.17 | 448.17 |
| 5 | 5e | 103 | 0 | 84.73 | −1.02 | 1.21 | 15.02 | 225.51 |
| 6 | 6a | 72 | 30.1 | 79.41 | −1.6 | 1.88 | 27.2 | 739.7 |
| 6 | 6b | 77 | 25.24 | 82.76 | −1.53 | 1.59 | 25.81 | 666.34 |
| 6 | 6c | 79 | 23.3 | 68.93 | −0.72 | 0.02 | 22.09 | 488.12 |
| 6 | 6d | 95 | 7.77 | 67.82 | −1.01 | 1.42 | 21.7 | 470.73 |
| 6 | 6e | 77 | 25.24 | 62.39 | −0.92 | 0.11 | 27.71 | 768.05 |
| 6 | 6f | 77 | 25.24 | 59.84 | −0.38 | −0.27 | 24.64 | 607.23 |
| 7 | 7a | 87 | 15.53 | 46.59 | −0.42 | 0.25 | 22.81 | 520.27 |
| 7 | 7b | 103 | 0 | 47.9 | −0.1 | 0.3 | 21.67 | 469.46 |
| 7 | 7c | 103 | 0 | 35.72 | 1.02 | −0.28 | 28.9 | 835.08 |
| 7 | 7d | 103 | 0 | 70.82 | −1.72 | 6.17 | 15.83 | 250.6 |
| 8 | 8a | 103 | 0 | 63.83 | −0.39 | −0.77 | 26.41 | 697.39 |
| 8 | 8b | 89 | 13.59 | 48.88 | −0.53 | −0.66 | 24.32 | 591.4 |
| 8 | 8c | 103 | 0 | 55.9 | −1.01 | 2.05 | 17.88 | 319.67 |
| 8 | 8d | 103 | 0 | 39.95 | −0.66 | 1.14 | 14.28 | 203.96 |
| 8 | 8e | 103 | 0 | 68.87 | −1.18 | 0.87 | 23.6 | 556.74 |
| 8 | 8f | 94 | 8.74 | 72.62 | −1.59 | 2.29 | 23.68 | 560.74 |
| 8 | 8g | 103 | 0 | 65.17 | −0.44 | 0.54 | 23.16 | 536.35 |
| 8 | 8h | 103 | 0 | 81.86 | −2.42 | 6.62 | 19.8 | 391.92 |
| 9 | 9a | 103 | 0 | 24.44 | 1.78 | 3.12 | 21.93 | 481.03 |
| 9 | 9b | 103 | 0 | 44.52 | 0.58 | −1.02 | 32.53 | 1058.11 |
| 9 | 9c | 94 | 8.74 | 37.18 | 0.76 | −0.29 | 26.2 | 686.43 |
| 9 | 9d | 101 | 1.94 | 12.26 | 2.28 | 5.16 | 20.05 | 402 |
| 9 | 9e | 103 | 0 | 23.75 | 1.25 | 1.21 | 23.21 | 538.58 |

**Table A2.** *Cont.*

| Goal | Indicator | Number of Cities | Missing Data (%) | Mean | Skewness | Kurtosis | Deviation | Variance |
|------|-----------|------------------|-------------------|------|----------|----------|-----------|----------|
| 9 | 9f | 95 | 7.77 | 32.61 | 1.07 | −0.11 | 31.89 | 1016.87 |
| 10 | 10a | 103 | 0 | 32.83 | 1.26 | 2.41 | 18.85 | 355.25 |
| 10 | 10b | 103 | 0 | 30.47 | 0.53 | 1.69 | 17.66 | 311.88 |
| 10 | 10e | 98 | 4.85 | 55.33 | −0.15 | 0.02 | 20.43 | 417.2 |
| 10 | 10d | 103 | 0 | 43.57 | 0.52 | 0.39 | 21.84 | 477.2 |
| 10 | 10e | 98 | 4.85 | 62.43 | −0.86 | 0.04 | 25.05 | 627.39 |
| 10 | 10f | 98 | 4.85 | 78.17 | −2.09 | 6.07 | 18.06 | 326.22 |
| 11 | 11a | 94 | 8.74 | 61.48 | −0.5 | −0.46 | 24.44 | 597.29 |
| 11 | 11b | 90 | 12.62 | 50.9 | 0.09 | −1.03 | 26.95 | 726.34 |
| 11 | 11c | 92 | 10.68 | 86.86 | −2.87 | 11.18 | 16.26 | 264.35 |
| 11 | 11d | 38 | 63.11 | 83.43 | −2.84 | 8.41 | 20.53 | 421.44 |
| 11 | 11e | 92 | 10.68 | 50.03 | −0.36 | −0.11 | 12.05 | 145.26 |
| 11 | 11f | 103 | 0 | 68.44 | −1.22 | 1.46 | 22.45 | 504.1 |
| 11 | 11g | 103 | 0 | 48.99 | 0.25 | −0.64 | 25.86 | 668.82 |
| 11 | 11h | 84 | 18.45 | 9.11 | 3.43 | 12.93 | 17.23 | 296.8 |
| 11 | 11i | 103 | 0 | 21.74 | 1.98 | 4.85 | 19.66 | 386.44 |
| 11 | 11j | 103 | 0 | 34.68 | 0.51 | −0.7 | 24.69 | 609.64 |
| 11 | 11k | 98 | 4.85 | 79.59 | −1.93 | 3.49 | 22.51 | 506.89 |
| 12 | 12a | 103 | 0 | 38.77 | 0.76 | −0.03 | 23.8 | 566.67 |
| 12 | 12b | 100 | 2.91 | 42.36 | 0.46 | 1.05 | 20.45 | 418.03 |
| 12 | 12c | 103 | 0 | 34.59 | 0.78 | 0.28 | 23.31 | 543.24 |
| 12 | 12d | 63 | 38.83 | 91.96 | −4.45 | 24.85 | 14.71 | 216.42 |
| 12 | 12e | 103 | 0 | 32.94 | 1.14 | 1.25 | 22.18 | 491.75 |
| 13 | 13a | 58 | 43.69 | 76.55 | −2.44 | 7.78 | 14.05 | 197.49 |
| 13 | 13b | 62 | 39.81 | 54.64 | −0.24 | −0.31 | 15 | 224.95 |
| 13 | 13c | 58 | 43.69 | 58.81 | −0.38 | 0.32 | 18.05 | 325.79 |
| 13 | 13d | 103 | 0 | 57.93 | −0.33 | −1.42 | 39.6 | 1567.79 |
| 14 | 14a | 45 | 56.31 | 41.18 | 0.17 | −1.26 | 34.52 | 1191.4 |
| 14 | 14b | 46 | 55.34 | 92.96 | −3.5 | 13.39 | 19.22 | 369.6 |
| 14 | 14c | 43 | 58.25 | 50.08 | −0.02 | −1.13 | 30.35 | 921.03 |
| 14 | 14d | 47 | 54.37 | 17.81 | 1.53 | 2.45 | 22.92 | 525.21 |
| 14 | 14e | 43 | 58.25 | 26.55 | 1.11 | 0 | 31.46 | 989.43 |
| 15 | 15a | 103 | 0 | 67.09 | −0.72 | −0.2 | 26.17 | 684.94 |
| 15 | 15b | 103 | 0 | 11.34 | 1.87 | 2.61 | 18.09 | 327.34 |
| 15 | 15c | 103 | 0 | 38.52 | 0.45 | −0.42 | 25.45 | 647.68 |
| 15 | 15d | 103 | 0 | 26.84 | 1.7 | 2.93 | 22.41 | 502.19 |
| 16 | 16a | 103 | 0 | 82.61 | −3.1 | 12.81 | 16.04 | 257.27 |
| 16 | 16b | 103 | 0 | 44.33 | −0.81 | 0.31 | 16.83 | 283.39 |
| 16 | 16c | 103 | 0 | 74.87 | −1.17 | 0.63 | 28.83 | 831.13 |
| 16 | 16d | 103 | 0 | 52.45 | −0.35 | −0.36 | 24.08 | 579.73 |
| 16 | 16e | 103 | 0 | 52.84 | −0.61 | −0.21 | 25.89 | 670.35 |
| 16 | 16f | 101 | 1.94 | 52.97 | −0.51 | 0.13 | 22.4 | 501.66 |
| 16 | 16g | 103 | 0 | 50.84 | −0.97 | 0.5 | 22.38 | 500.78 |
| 16 | 16h | 103 | 0 | 56.96 | −0.5 | −0.66 | 29.88 | 892.78 |
| 16 | 16i | 85 | 17.48 | 50.82 | −1.96 | 6.31 | 12.33 | 152.08 |
| 17 | 17a | 103 | 0 | 13.98 | 2.19 | 4.37 | 23.22 | 539 |
| 17 | 17b | 103 | 0 | 32.52 | 0.76 | −1.41 | 46.29 | 2142.59 |
| 17 | 17c | 103 | 0 | 41.05 | 0.42 | −0.32 | 23.71 | 562.08 |
| 17 | 17d | 103 | 0 | 86.58 | −2.25 | 4.68 | 22.66 | 513.64 |

## Appendix C

**Table A3.** Normalized values used in the alternative weighting.

| Goal | Assessment Value | Normalized Value |
|------|------------------|------------------|
| 1 | 2.08 | 1.09 |
| 2 | 1.80 | 0.94 |
| 3 | 1.72 | 0.89 |
| 4 | 1.54 | 0.80 |
| 5 | 2.29 | 1.20 |
| 6 | 2.80 | 1.47 |
| 7 | 1.76 | 0.91 |
| 8 | 1.95 | 1.02 |
| 9 | 1.97 | 1.03 |
| 10 | 2.28 | 1.19 |
| 11 | 2.85 | 1.50 |
| 12 | 2.29 | 1.20 |
| 13 | 2.18 | 1.14 |
| 14 | 0.99 | 0.50 |
| 15 | 1.38 | 0.71 |
| 16 | 2.18 | 1.14 |
| 17 | 2.35 | 1.23 |

**Table A4.** Correlations between the indicators, their respective goal and the overall index. Numbers represent the Pearson correlation coefficients between each indicator and the corresponding goal and between each indicator and the overall index. Correlations that are not significant at the significance level of $\alpha = 0.01$ are in grey. Very high correlations (i.e., Pearson correlation coefficients greater than 0.70) are bolded and negative correlations highlighted in red.

| Id | Respective SDG | | General Index | |
|----|--------------|-----------|--------------|-----------|
| | Coefficients | Indicator | Coefficients | Indicator |
| 1a | 0.51 | 0.00 | 0.01 | 0.92 |
| 1b | 0.27 | 0.01 | 0.04 | 0.70 |
| 1c | **0.90** | 0.00 | 0.66 | 0.00 |
| 1d | **0.87** | 0.00 | 0.64 | 0.00 |
| 1e | **0.92** | 0.00 | 0.65 | 0.00 |
| 2a | 0.39 | 0.00 | 0.04 | 0.63 |
| 2b | 0.43 | 0.00 | 0.05 | 0.65 |
| 2c | 0.66 | 0.00 | 0.02 | 0.83 |
| 2d | **0.78** | 0.00 | 0.16 | 0.11 |
| 2e | **0.76** | 0.00 | 0.12 | 0.25 |
| 3a | 0.28 | 0.00 | 0.03 | 0.80 |
| 3b | 0.46 | 0.00 | 0.10 | 0.32 |
| 3c | **0.72** | 0.00 | −0.07 | 0.49 |
| 3d | 0.58 | 0.00 | 0.05 | 0.59 |
| 3e | 0.09 | 0.40 | 0.10 | 0.36 |
| 3f | **0.90** | 0.00 | −0.10 | 0.32 |
| 3g | **0.92** | 0.00 | 0.01 | 0.96 |
| 3h | 0.56 | 0.00 | −0.01 | 0.93 |
| 3i | 0.10 | 0.31 | 0.07 | 0.46 |
| 3j | 0.56 | 0.00 | −0.11 | 0.26 |
| 3k | **0.88** | 0.00 | −0.06 | 0.56 |
| 3l | 0.23 | 0.02 | 0.51 | 0.00 |
| 3m | 0.67 | 0.00 | 0.07 | 0.45 |
| 4a | 0.58 | 0.00 | 0.29 | 0.01 |
| 4b | 0.49 | 0.00 | 0.14 | 0.18 |
| 4c | 0.62 | 0.00 | 0.22 | 0.03 |
| 4d | **0.83** | 0.00 | 0.53 | 0.00 |

**Table A4.** *Cont.*

| Id | Respective SDG | | General Index | |
|---|---|---|---|---|
| | Coefficients | Indicator | Coefficients | Indicator |
| 4e | −0.05 | 0.65 | −0.06 | 0.56 |
| 4f | **0.82** | 0.00 | 0.53 | 0.00 |
| 5a | 0.42 | 0.00 | −0.01 | 0.95 |
| 5b | 0.38 | 0.00 | −0.10 | 0.32 |
| 5c | 0.46 | 0.00 | 0.26 | 0.01 |
| 5d | 0.58 | 0.00 | 0.28 | 0.01 |
| 5e | 0.51 | 0.00 | 0.19 | 0.05 |
| 6a | 0.49 | 0.00 | −0.01 | 0.91 |
| 6b | 0.35 | 0.00 | 0.06 | 0.58 |
| 6c | 0.67 | 0.00 | 0.41 | 0.00 |
| 6d | 0.46 | 0.00 | 0.11 | 0.29 |
| 6e | 0.53 | 0.00 | 0.41 | 0.00 |
| 6f | 0.18 | 0.12 | −0.02 | 0.85 |
| 7a | 0.40 | 0.00 | 0.18 | 0.09 |
| 7b | 0.48 | 0.00 | 0.43 | 0.00 |
| 7c | 0.58 | 0.00 | 0.23 | 0.02 |
| 7d | 0.63 | 0.00 | 0.39 | 0.00 |
| 8a | 0.64 | 0.00 | 0.25 | 0.01 |
| 8b | 0.61 | 0.00 | 0.48 | 0.00 |
| 8c | 0.21 | 0.03 | 0.19 | 0.05 |
| 8d | 0.26 | 0.01 | 0.12 | 0.21 |
| 8e | 0.61 | 0.00 | 0.46 | 0.00 |
| 8f | 0.37 | 0.00 | −0.03 | 0.75 |
| 8g | 0.23 | 0.02 | −0.26 | 0.01 |
| 8h | 0.48 | 0.00 | 0.15 | 0.14 |
| 9a | 0.06 | 0.53 | 0.22 | 0.03 |
| 9b | 0.38 | 0.00 | 0.03 | 0.77 |
| 9c | 0.56 | 0.00 | 0.22 | 0.03 |
| 9d | 0.23 | 0.02 | −0.22 | 0.03 |
| 9e | 0.33 | 0.00 | 0.35 | 0.00 |
| 9f | 0.52 | 0.00 | 0.01 | 0.90 |
| 10a | 0.41 | 0.00 | 0.12 | 0.21 |
| 10b | 0.50 | 0.00 | 0.21 | 0.03 |
| 10e | **0.82** | 0.00 | 0.07 | 0.52 |
| 10d | 0.14 | 0.17 | −0.26 | 0.01 |
| 10e | 0.54 | 0.00 | 0.66 | 0.00 |
| 10f | 0.53 | 0.00 | −0.07 | 0.47 |
| 11a | 0.37 | 0.00 | −0.05 | 0.63 |
| 11b | 0.35 | 0.00 | 0.13 | 0.23 |
| 11c | 0.40 | 0.00 | 0.33 | 0.00 |
| 11d | 0.35 | 0.03 | 0.24 | 0.15 |
| 11e | 0.36 | 0.00 | 0.38 | 0.00 |
| 11f | 0.44 | 0.00 | 0.20 | 0.05 |
| 11g | 0.18 | 0.06 | −0.15 | 0.14 |
| 11h | 0.17 | 0.12 | 0.15 | 0.19 |
| 11i | −0.004 | 0.97 | 0.01 | 0.94 |
| 11j | 0.30 | 0.00 | 0.15 | 0.12 |
| 11k | 0.24 | 0.02 | 0.09 | 0.40 |
| 12a | 0.60 | 0.00 | 0.33 | 0.00 |
| 12b | 0.37 | 0.00 | 0.14 | 0.16 |
| 12c | **0.74** | 0.00 | 0.46 | 0.00 |
| 12d | 0.11 | 0.37 | 0.38 | 0.00 |
| 12e | 0.70 | 0.00 | 0.11 | 0.27 |
| 13a | 0.59 | 0.00 | −0.12 | 0.37 |
| 13b | **0.77** | 0.00 | 0.17 | 0.18 |
| 13c | 0.65 | 0.00 | 0.35 | 0.01 |
| 13d | **0.95** | 0.00 | 0.30 | 0.00 |

**Table A4.** *Cont.*

| Id | Respective SDG | | General Index | |
| | Coefficients | Indicator | Coefficients | Indicator |
|---|---|---|---|---|
| 14a | 0.56 | 0.00 | −0.19 | 0.21 |
| 14b | 0.61 | 0.00 | −0.13 | 0.38 |
| 14c | 0.70 | 0.00 | 0.21 | 0.19 |
| 14d | 0.61 | 0.00 | 0.08 | 0.60 |
| 14e | **0.84** | 0.00 | 0.25 | 0.11 |
| 15a | **0.72** | 0.00 | 0.13 | 0.20 |
| 15b | 0.48 | 0.00 | 0.03 | 0.74 |
| 15c | **0.78** | 0.00 | 0.09 | 0.37 |
| 15d | 0.51 | 0.00 | 0.17 | 0.09 |
| 16a | 0.48 | 0.00 | 0.49 | 0.00 |
| 16b | 0.35 | 0.00 | 0.27 | 0.01 |
| 16c | 0.52 | 0.00 | 0.44 | 0.00 |
| 16d | 0.53 | 0.00 | 0.55 | 0.00 |
| 16e | 0.68 | 0.00 | 0.25 | 0.01 |
| 16f | 0.36 | 0.00 | 0.09 | 0.36 |
| 16g | **0.72** | 0.00 | 0.21 | 0.03 |
| 16h | 0.66 | 0.00 | 0.18 | 0.06 |
| 16i | 0.64 | 0.00 | 0.43 | 0.00 |
| 17a | 0.67 | 0.00 | 0.51 | 0.00 |
| 17b | **0.82** | 0.00 | 0.38 | 0.00 |
| 17c | **0.72** | 0.00 | 0.56 | 0.00 |
| 17d | 0.48 | 0.00 | 0.14 | 0.16 |

**Table A5.** Correlations between the goals and the overall index. Numbers represent the Pearson correlation coefficients between the SDG goals and the overall index. Significant correlations greater than 0.01 are in grey. High positive correlations are highlighted in green and negative in red.

| SDG | | Index | 1 | 2 | 3 | 4 | 5 | 6 | 7 | 8 | 9 | 10 | 11 | 12 | 13 | 14 | 15 | 16 | 17 |
|---|---|---|---|---|---|---|---|---|---|---|---|---|---|---|---|---|---|---|---|
| Index | Pearson's corr. | 1.00 | | | | | | | | | | | | | | | | | |
| | Significance coef. | | | | | | | | | | | | | | | | | | |
| Goal 1 | Pearson's corr. | 0.54 | 1.00 | | | | | | | | | | | | | | | | |
| | Significance coef. | 0.00 | | | | | | | | | | | | | | | | | |
| Goal 2 | Pearson's corr. | 0.12 | −0.21 | 1.00 | | | | | | | | | | | | | | | |
| | Significance coef. | 0.25 | 0.04 | | | | | | | | | | | | | | | | |
| Goal 3 | Pearson's corr. | 0.01 | 0.01 | −0.15 | 1.00 | | | | | | | | | | | | | | |
| | Significance coef. | 0.91 | 0.93 | 0.14 | | | | | | | | | | | | | | | |
| Goal 4 | Pearson's corr. | 0.47 | 0.58 | −0.23 | −0.14 | 1.00 | | | | | | | | | | | | | |
| | Significance coef. | 0.00 | 0.00 | 0.02 | 0.17 | | | | | | | | | | | | | | |
| Goal 5 | Pearson's corr. | 0.35 | 0.13 | 0.09 | −0.21 | 0.09 | 1.00 | | | | | | | | | | | | |
| | Significance coef. | 0.00 | 0.20 | 0.38 | 0.04 | 0.38 | | | | | | | | | | | | | |
| Goal 6 | Pearson's corr. | 0.27 | 0.16 | 0.10 | 0.01 | −0.01 | 0.19 | 1.00 | | | | | | | | | | | |
| | Significance coef. | 0.01 | 0.12 | 0.31 | 0.95 | 0.91 | 0.05 | | | | | | | | | | | | |
| Goal 7 | Pearson's corr. | 0.58 | 0.43 | −0.02 | −0.16 | 0.42 | 0.34 | 0.31 | 1.00 | | | | | | | | | | |
| | Significance coef. | 0.00 | 0.00 | 0.87 | 0.10 | 0.00 | 0.00 | 0.00 | | | | | | | | | | | |
| Goal 8 | Pearson's corr. | 0.38 | 0.42 | −0.15 | 0.04 | 0.19 | −0.05 | 0.24 | 0.31 | 1.00 | | | | | | | | | |
| | Significance coef. | 0.00 | 0.00 | 0.14 | 0.70 | 0.05 | 0.60 | 0.01 | 0.00 | | | | | | | | | | |
| Goal 9 | Pearson's corr. | 0.31 | 0.38 | 0.04 | 0.26 | 0.10 | −0.06 | 0.05 | 0.05 | 0.26 | 1.00 | | | | | | | | |
| | Significance coef. | 0.00 | 0.00 | 0.68 | 0.01 | 0.31 | 0.57 | 0.58 | 0.27 | 0.01 | | | | | | | | | |
| Goal 10 | Pearson's corr. | 0.19 | 0.47 | 0.20 | 0.07 | −0.02 | −0.01 | 0.09 | 0.14 | 0.23 | 0.34 | 1.00 | | | | | | | |
| | Significance coef. | 0.05 | 0.00 | 0.05 | 0.51 | 0.88 | 0.90 | 0.36 | 0.17 | 0.02 | 0.00 | | | | | | | | |
| Goal 11 | Pearson's corr. | 0.29 | −0.02 | 0.36 | −0.16 | 0.06 | 0.12 | 0.11 | 0.10 | −0.13 | −0.07 | 0.11 | 1.00 | | | | | | |
| | Significance coef. | 0.00 | 0.85 | 0.00 | 0.10 | 0.55 | 0.22 | 0.29 | 0.31 | 0.18 | 0.48 | 0.25 | | | | | | | |
| Goal 12 | Pearson's corr. | 0.47 | 0.26 | −0.11 | −0.14 | 0.39 | 0.12 | −0.16 | 0.31 | 0.05 | 0.10 | −0.18 | −0.12 | 1.00 | | | | | |
| | Significance coef. | 0.00 | 0.01 | 0.28 | 0.15 | 0.00 | 0.25 | 0.10 | 0.00 | 0.65 | 0.29 | 0.06 | 0.21 | | | | | | |
| Goal 13 | Pearson's corr. | 0.37 | −0.01 | −0.24 | −0.01 | 0.05 | 0.04 | −0.17 | −0.18 | 0.04 | −0.01 | −0.19 | 0.10 | 0.16 | 1.00 | | | | |
| | Significance coef. | 0.00 | 0.89 | 0.01 | 0.91 | 0.65 | 0.68 | 0.09 | 0.07 | 0.69 | 0.93 | 0.06 | 0.33 | 0.11 | | | | | |
| Goal 14 | Pearson's corr. | 0.08 | −0.38 | 0.28 | 0.04 | −0.30 | 0.02 | 0.05 | −0.14 | −0.22 | −0.16 | 0.04 | 0.27 | −0.21 | 0.05 | 1.00 | | | |
| | Significance coef. | 0.43 | 0.00 | 0.00 | 0.68 | 0.00 | 0.81 | 0.17 | 0.03 | 0.10 | 0.66 | 0.01 | 0.03 | 0.62 | | | | | |
| Goal 15 | Pearson's corr. | 0.17 | −0.21 | 0.24 | −0.19 | −0.10 | 0.27 | −0.13 | 0.18 | −0.22 | −0.20 | −0.10 | 0.12 | 0.03 | −0.02 | 0.31 | 1.00 | | |
| | Significance coef. | 0.09 | 0.03 | 0.01 | 0.06 | 0.31 | 0.01 | 0.18 | 0.07 | 0.03 | 0.04 | 0.32 | 0.25 | 0.74 | 0.81 | 0.00 | | | |
| Goal 16 | Pearson's corr. | 0.57 | 0.31 | 0.12 | −0.14 | 0.35 | 0.23 | 0.25 | 0.56 | 0.37 | 0.03 | −0.01 | −0.01 | 0.30 | −0.12 | −0.03 | 0.13 | 1.00 | |
| | Significance coef. | 0.00 | 0.00 | 0.24 | 0.17 | 0.00 | 0.02 | 0.01 | 0.00 | 0.00 | 0.74 | 0.94 | 0.90 | 0.00 | 0.21 | 0.78 | 0.20 | | |
| Goal 17 | Pearson's corr. | 0.56 | 0.32 | −0.26 | −0.04 | 0.42 | 0.03 | 0.04 | 0.28 | 0.22 | 0.06 | −0.22 | 0.08 | 0.53 | 0.32 | −0.30 | −0.17 | 0.30 | 1.00 |
| | Significance coef. | 0.00 | 0.00 | 0.01 | 0.70 | 0.00 | 0.75 | 0.66 | 0.00 | 0.02 | 0.54 | 0.03 | 0.44 | 0.00 | 0.00 | 0.00 | 0.08 | 0.00 | |

**Table A6.** Summary of main correlations between the indicators.

| Indicators Id | Pearson Coef. | Indicators Id | Pearson Coef. | Indicators Id | Pearson Coef. | Indicators Id | Pearson Coef. |
|---|---|---|---|---|---|---|---|
| 15a/11j | −0.78 | 16h/1e | 0.89 | 3g/3d | 0.77 | 7b/1d | 0.74 |
| 10a/1e | 0.99 | 8b/8e | 0.81 | 3m/1d | 0.77 | 7b/1c | 0.74 |
| 10a/1c | 0.99 | 16d/3m | 0.81 | 6e/6a | 0.77 | 8c/1d | 0.74 |
| 1c/1e | 0.98 | 10d/1a | 0.8 | 16d/1d | 0.77 | 3m/1e | 0.73 |
| 1c/1d | 0.97 | 16h/16e | 0.8 | 16d/10a | 0.77 | 4g/1d | 0.73 |
| 10a/1d | 0.96 | 16g/16h | 0.79 | 16d/1e | 0.76 | 8c/1e | 0.73 |
| 1d/1c | 0.95 | 4d/1d | 0.78 | 16g/1e | 0.76 | 10f/10d | 0.73 |
| 4d/4g | 0.95 | 5a/5b | 0.78 | 3m/1c | 0.75 | 8e/3m | 0.71 |
| 3g/3l | 0.92 | 8c/1c | 0.78 | 10a/3m | 0.75 | 11c/11d | 0.71 |
| 3h/3i | 0.91 | 8b/3m | 0.78 | 10a/8b | 0.75 | 16a/1c | 0.71 |
| 3h/3g | 0.91 | 16d/1c | 0.78 | 4d/1c | 0.74 | 4d/1c | 0.69 |

**Table A7.** Total explained variance for $I_s$ and $A_s$.

| Component | Initial Set ($I_s$) | | | Alternative Set ($A_s$) | | |
|---|---|---|---|---|---|---|
| | Eigenvalue | % Variance | % Accumulated | Eigenvalue | % Variance | % Accumulated |
| 1 | 3.634 | 21.37 | 21.37 | 3.183 | 18.72 | 18.72 |
| 2 | 2.238 | 13.16 | 34.54 | 2.234 | 13.14 | 18.72 |
| 3 | 2.068 | 12.16 | 46.70 | 1.854 | 10.90 | 31.86 |
| 4 | 1.280 | 7.53 | 54.23 | 1.442 | 8.48 | 42.77 |
| 5 | 1.147 | 6.74 | 60.98 | 1.151 | 6.77 | 51.25 |
| 6 | 1.013 | 5.95 | 66.93 | 1.131 | 6.65 | 58.02 |
| 7 | 0.904 | 5.31 | 72.25 | 1.088 | 6.40 | 64.67 |
| 8 | 0.852 | 5.01 | 77.20 | 0.870 | 5.12 | 71.07 |
| 9 | 0.756 | 4.44 | 81.71 | 0.751 | 4.42 | 76.19 |
| 10 | 0.553 | 3.25 | 84.96 | 0.623 | 3.66 | 80.61 |
| 11 | 0.548 | 3.22 | 88.18 | 0.551 | 3.24 | 84.28 |
| 12 | 0.484 | 2.85 | 91.03 | 0.545 | 3.20 | 87.52 |
| 13 | 0.435 | 2.55 | 93.59 | 0.477 | 2.80 | 90.73 |
| 14 | 0.324 | 1.90 | 95.50 | 0.316 | 1.85 | 93.53 |
| 15 | 0.320 | 1.88 | 97.38 | 0.287 | 1.68 | 95.39 |
| 16 | 0.256 | 1.50 | 98.89 | 0.270 | 1.59 | 97.08 |
| 17 | 0.188 | 1.10 | 100 | 0.225 | 1.32 | 98.67 |

**Table A8.** Rotated Component Matrix for $I_s$ and $A_s$.

| SDG | Initial Set ($I_s$) | | | | Alternative Set ($A_s$) | | | |
|---|---|---|---|---|---|---|---|---|
| | 1 | 2 | 3 | 4 | 1 | 2 | 3 | 4 |
| 1 | | 0.787 | | | | −0.674 | 0.471 | |
| 2 | 0.698 | | | | | | | |
| 3 | | | | | | | | |
| 4 | −0.309 | 0.442 | | 0.385 | 0.567 | −0.363 | | |
| 5 | | | 0.616 | | | | 0.742 | |
| 6 | | | | | | | | 0.708 |
| 7 | | 0.310 | 0.678 | | 0.420 | | 0.371 | |
| 8 | | 0.330 | | | | | | 0.763 |
| 9 | | 0.590 | | | | | 0.336 | |
| 10 | | 0.771 | | −0.332 | | | 0.565 | 0.392 |
| 11 | 0.734 | | | | | | 0.308 | |
| 12 | −0.426 | | 0.470 | 0.452 | 0.797 | | | −0.304 |
| 13 | | | | 0.765 | | | | |
| 14 | 0.698 | | | | −0.317 | 0.731 | | |
| 15 | 0.308 | | 0.648 | | | 0.697 | | |
| 16 | | | 0.652 | | 0.603 | | | 0.497 |
| 17 | | | | 0.770 | 0.721 | | | |

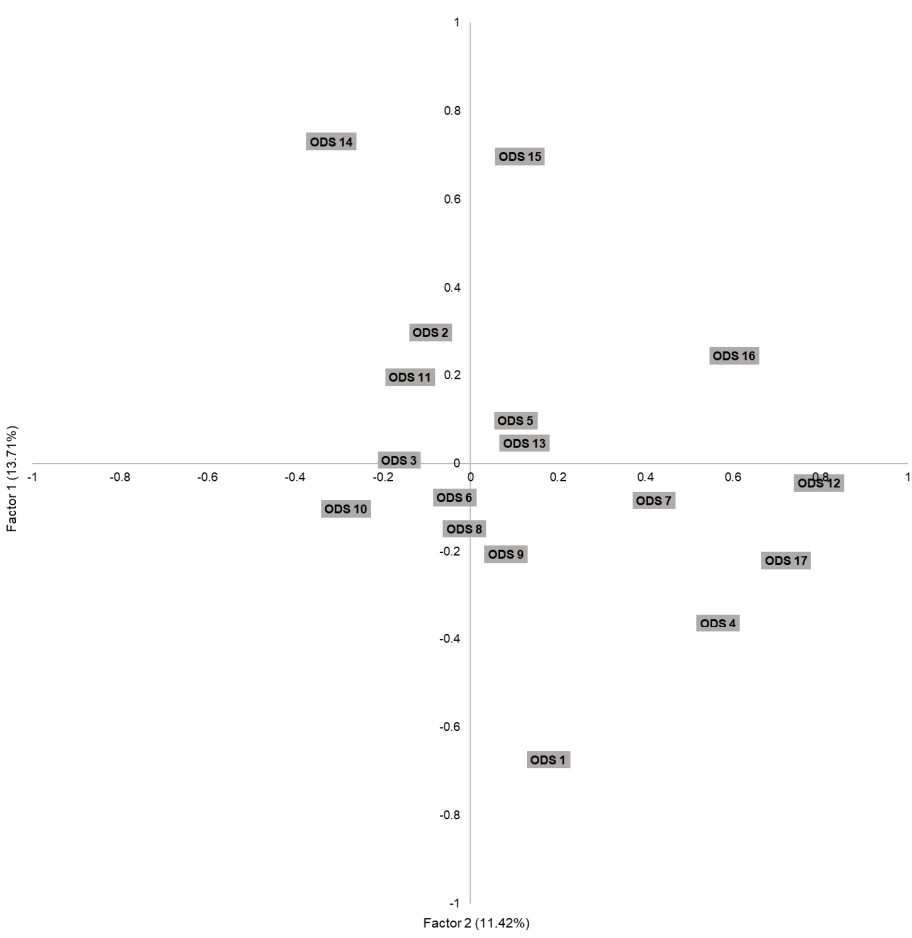

**Figure A1.** Factor map of the 17 goals of the SCR Index for $A_s$.

**Table A9.** Total explained variance of the intermediate indices. In bold values with total explained variance above 60%.

| SDG | Initial Set ($I_s$) | | | Alternative Set ($A_s$) | | | Variance Difference |
|---|---|---|---|---|---|---|---|
| | Factors | Indicators | % Variance | Factors | Indicators | % Variance | |
| 1 | 1 | 5 | **60.75** | 1 | 3 | 43.82 | −27.87 |
| 2 | 2 | 5 | **64.00** | 1 | 3 | 63.56 | −0.69 |
| 3 | 4 | 13 | **66.49** | 4 | 12 | **64.48** | −3.02 |
| 4 | 2 | 6 | **66.95** | 2 | 5 | **65.84** | −1.66 |
| 5 | 2 | 5 | **64.43** | 1 | 3 | 46.63 | −27.63 |
| 6 | 3 | 6 | **69.82** | 2 | 4 | 55.27 | −20.84 |
| 7 | 2 | 4 | **61.97** | 1 | 2 | 50.58 | −18.38 |
| 8 | 3 | 8 | **63.90** | 3 | 6 | **68.04** | 6.48 |
| 9 | 3 | 6 | **69.03** | 2 | 4 | 59.89 | −13.24 |
| 10 | 3 | 6 | **77.13** | 1 | 3 | 50.82 | −34.11 |
| 11 | 4 | 11 | **61.56** | 3 | 8 | **61.49** | −0.11 |
| 12 | 2 | 5 | **67.04** | | | | |
| 13 | 2 | 4 | **66.83** | | | | |
| 14 | 1 | 5 | 44.54 | | | | |
| 15 | 1 | 4 | 41.13 | | | | |
| 16 | 3 | 9 | **69.50** | 2 | 7 | 57.81 | −16.82 |
| 17 | 1 | 4 | 48.49 | | | | |

**Table A10.** Rotated component matrix for SDG 3. In bold values greater than 0.700.

| Indicators | 1 | 2 | 3 | 4 |
|---|---|---|---|---|
| n_sdg03_adfertility | | **0.794** | | |
| n_sdg03_alcohol | 0.432 | | | |
| n_sdg03_gripe | 0.668 | | −0.308 | 0.365 |
| n_sdg03_hepatitis | 0.672 | | 0.323 | |
| n_sdg03_infantil | | | **0.703** | |
| n_sdg03_prematuras | **0.891** | | | |
| n_sdg03_suicidios | 0.403 | 0.486 | | |
| n_sdg03_trafico | | | | **0.839** |
| n_sdg03_tuberculosis | 0.339 | 0.441 | −0.383 | 0.404 |
| n_sdg03_tumores | **0.883** | | | |
| n_sdg03_vida | | 0.595 | 0.594 | |
| n_sdg03_vih | **0.811** | | | |

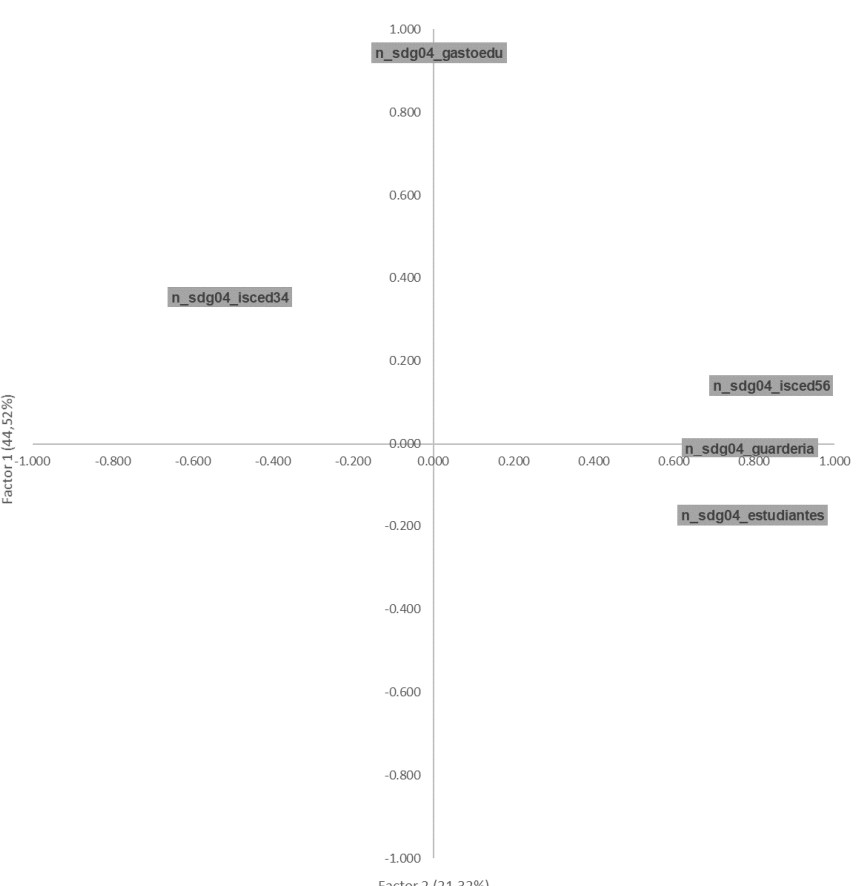

**Figure A2.** Rotated space component for SDG4.

**Table A11.** Rotated component matrix for SDG 8. In bold values greater than 0.700.

| Indicators | 1 | 2 | 3 |
|---|---|---|---|
| n_sdg08_desempleo | | 0.762 | |
| n_sdg08_desempleocovid | | | 0.833 |
| n_sdg08_desempleojovenes | | 0.761 | |
| n_sdg08_diversidad | **0.710** | −0.327 | |
| n_sdg08_pibcapitamun | 0.454 | | −0.663 |
| n_sdg08_productividad | **0.821** | | |

**Table A12.** Rotated component matrix for SDG 11. In bold values greater than 0.700.

| Indicators | 1 | 2 | 3 |
|---|---|---|---|
| n_sdg11_no2 | **0.76** | | |
| n_sdg11_o3 | | | **−0.77** |
| n_sdg11_pm10 | | **0.88** | |
| n_sdg11_pm10media | | **0.85** | |
| n_sdg11_preciovivienda | 0.64 | | |
| n_sdg11_resiliencia | 0.59 | | 0.49 |
| n_sdg11_suptrans | **−0.79** | | |
| n_sdg11_vulnerables | | | 0.55 |

## Appendix C

**Table A13.** Top 10 positions of cities on each calculation alternative.

| Position | Initial Set ($I_s$) | | Alternative Set ($A_s$) | |
|---|---|---|---|---|
| | City | Index Score | City | Index Score |
| 1 | Vitoria-Gasteiz | 61.05 | Vitoria-Gasteiz | 68.33 |
| 2 | Zaragoza | 58.72 | Zaragoza | 66.60 |
| 3 | Logroño | 57.23 | Logroño | 63.60 |
| 4 | Soria | 56.15 | Getafe | 62.53 |
| 5 | Getafe | 55.04 | Soria | 61.31 |
| 6 | Lleida | 53.47 | Burgos | 60.90 |
| 7 | Palencia | 53.47 | Rivas-Vaciamadrid | 60.73 |
| 8 | Cáceres | 53.46 | Palencia | 60.72 |
| 9 | Madrid | 53.36 | Cáceres | 59.98 |
| 10 | Donostia-San Sebastián | 53.19 | Móstoles | 59.93 |

**Table A14.** Bottom 10 positions of cities on each calculation alternative.

| Position | Initial Set ($I_s$) | | Alternative Set ($A_s$) | |
|---|---|---|---|---|
| | City | Index Score | City | Index Score |
| 1 | Barakaldo | 41.11 | Teruel | 45.13 |
| 2 | Arona | 41.03 | Barakaldo | 44.69 |
| 3 | Talavera de la Reina | 40.60 | Melilla | 44.63 |
| 4 | Melilla | 40.47 | Vélez-Málaga | 42.74 |
| 5 | Marbella | 39.86 | Arona | 42.06 |
| 6 | Vélez-Málaga | 39.82 | El Ejido | 41.85 |
| 7 | El Ejido | 38.67 | Marbella | 41.19 |
| 8 | Fuengirola | 38.07 | Ceuta | 40.35 |
| 9 | Ceuta | 35.75 | Fuengirola | 40.04 |
| 10 | Torrevieja | 35.50 | Torrevieja | 36.60 |

**Table A15.** Full list of cities and index score on each calculation variation.

| City | Initial Set ($I_s$) | | Alternative Set ($A_s$) | | Shifts |
|---|---|---|---|---|---|
| | Position | Index Score | Position | Index Score | |
| Albacete | 44 | 48.96 | 31 | 56.61 | 13 |
| Alcalá de Henares | 28 | 50.95 | 22 | 58.09 | 6 |
| Alcobendas | 27 | 51.06 | 25 | 57.83 | 2 |
| Alcorcón | 20 | 51.72 | 14 | 59.08 | 6 |
| Algeciras | 85 | 43.34 | 91 | 45.68 | −6 |
| Alicante | 64 | 46.47 | 78 | 48.62 | −14 |
| Almería | 83 | 43.56 | 86 | 47.12 | −3 |
| Arona | 95 | 41.03 | 98 | 42.06 | −3 |
| Ávila | 24 | 51.51 | 26 | 57.23 | −2 |
| Avilés | 89 | 42.69 | 81 | 48.38 | 8 |

**Table A15.** *Cont.*

| City | Initial Set (I$_s$) | | Alternative Set (A$_s$) | | |
|---|---|---|---|---|---|
| | Position | Index Score | Position | Index Score | Shifts |
| Badajoz | 59 | 47.29 | 62 | 51.77 | −3 |
| Badalona | 77 | 44.35 | 66 | 51.27 | 11 |
| Barakaldo | 94 | 41.11 | 95 | 44.69 | −1 |
| Barcelona | 23 | 51.63 | 23 | 57.88 | 0 |
| Bilbao | 43 | 49.10 | 41 | 54.87 | 2 |
| Burgos | 15 | 52.63 | 6 | 60.90 | 9 |
| Cáceres | 8 | 53.46 | 9 | 59.98 | −1 |
| Cádiz | 58 | 47.30 | 47 | 53.53 | 11 |
| Cartagena | 52 | 47.82 | 56 | 52.13 | −4 |
| Castellón de la Planta | 53 | 47.64 | 49 | 53.22 | 4 |
| Ceuta | 102 | 35.75 | 101 | 40.35 | 1 |
| Chiclana de la Frontera | 61 | 46.83 | 80 | 48.46 | −19 |
| Ciudad Real | 66 | 46.22 | 55 | 52.64 | 11 |
| Córdoba | 22 | 51.67 | 19 | 58.66 | 3 |
| Cornellá de Llobregat | 42 | 49.23 | 16 | 58.90 | 26 |
| Coslada | 88 | 42.93 | 75 | 49.02 | 13 |
| Cuenca | 16 | 52.02 | 44 | 54.38 | −28 |
| Donostia-San Sebastián | 10 | 53.19 | 45 | 54.05 | −35 |
| Dos Hermanas | 90 | 42.45 | 88 | 46.78 | 2 |
| El Ejido | 100 | 38.67 | 99 | 41.85 | 1 |
| El Puerto de Santa María | 51 | 47.92 | 67 | 51.23 | −16 |
| Elche | 46 | 48.57 | 58 | 51.92 | −12 |
| Fuengirola | 101 | 38.07 | 102 | 40.04 | −1 |
| Fuenlabrada | 49 | 48.24 | 34 | 56.14 | 15 |
| Getafe | 5 | 55.04 | 4 | 62.53 | 1 |
| Gijón | 14 | 52.63 | 18 | 58.70 | −4 |
| Girona | 12 | 53.06 | 12 | 59.23 | 0 |
| Granada | 65 | 46.31 | 65 | 51.30 | 0 |
| Guadalajara | 38 | 49.85 | 28 | 56.82 | 10 |
| L'Hospitalet de Llobregat | 93 | 41.81 | 83 | 48.18 | 10 |
| Huelva | 80 | 44.15 | 84 | 47.45 | −4 |
| Huesca | 26 | 51.33 | 35 | 56.01 | −9 |
| Jaén | 62 | 46.55 | 46 | 53.66 | 16 |
| Jerez de la Frontera | 79 | 44.23 | 82 | 48.28 | −3 |
| A Coruña | 39 | 49.72 | 43 | 54.63 | −4 |
| Las Palmas de GC | 68 | 45.47 | 59 | 51.82 | 9 |
| Las Rozas de Madrid | 36 | 49.89 | 40 | 55.14 | −4 |
| Leganés | 70 | 45.04 | 64 | 51.54 | 6 |
| León | 35 | 50.09 | 30 | 56.72 | 5 |
| Lleida | 6 | 53.47 | 17 | 58.82 | −11 |
| Logroño | 3 | 57.23 | 3 | 63.60 | 0 |
| Lorca | 13 | 52.76 | 13 | 59.09 | 0 |
| Lugo | 31 | 50.62 | 52 | 52.87 | −21 |
| Madrid | 9 | 53.36 | 11 | 59.77 | −2 |
| Málaga | 54 | 47.63 | 51 | 53.01 | 3 |
| Marbella | 98 | 39.86 | 100 | 41.19 | −2 |
| Mataró | 72 | 44.85 | 74 | 49.36 | −2 |
| Melilla | 97 | 40.47 | 96 | 44.63 | 1 |
| Mérida | 75 | 44.66 | 87 | 46.90 | −12 |
| Mijas | 69 | 45.45 | 90 | 46.62 | −21 |
| Móstoles | 17 | 51.88 | 10 | 59.93 | 7 |
| Murcia | 60 | 46.92 | 63 | 51.66 | −3 |
| Ourense | 57 | 47.34 | 71 | 50.09 | −14 |
| Oviedo | 41 | 49.49 | 32 | 56.61 | 9 |
| Palencia | 7 | 53.47 | 8 | 60.72 | −1 |
| Palma de Mallorca | 71 | 44.87 | 85 | 47.30 | −14 |
| Pamplona | 25 | 51.40 | 38 | 55.50 | −13 |

**Table A15.** *Cont.*

| City | Initial Set (I$_s$) | | Alternative Set (A$_s$) | | |
| | Position | Index Score | Position | Index Score | Shifts |
|---|---|---|---|---|---|
| Parla | 78 | 44.31 | 68 | 50.89 | 10 |
| Pontevedra | 67 | 45.68 | 77 | 48.93 | −10 |
| Pozuelo de Alarcón | 29 | 50.90 | 37 | 55.55 | −8 |
| Reus | 81 | 44.05 | 70 | 50.53 | 11 |
| Rivas-Vaciamadrid | 11 | 53.08 | 7 | 60.73 | 4 |
| Roquetas de Mar | 92 | 41.98 | 93 | 45.23 | −1 |
| Sabadell | 40 | 49.61 | 15 | 59.08 | 25 |
| Salamanca | 55 | 47.60 | 48 | 53.46 | 7 |
| San Boi de Llobregat | 56 | 47.48 | 29 | 56.82 | 27 |
| San Cristobal La Laguna | 50 | 48.04 | 61 | 51.78 | −11 |
| Sant Cugat del Vallès | 18 | 51.76 | 21 | 58.12 | −3 |
| San Fernando | 47 | 48.40 | 57 | 52.10 | −10 |
| San Sebastián de los Reyes | 63 | 46.51 | 54 | 52.75 | 9 |
| Santa Coloma de Gramenet | 87 | 43.05 | 76 | 48.97 | 11 |
| Santa Cruz de Tenerife | 32 | 50.35 | 39 | 55.49 | −7 |
| Santander | 21 | 51.68 | 27 | 57.21 | −6 |
| Santiago de Compostela | 37 | 49.85 | 33 | 56.36 | 4 |
| Segovia | 76 | 44.62 | 73 | 49.52 | 3 |
| Sevilla | 82 | 43.81 | 79 | 48.61 | 3 |
| Soria | 4 | 56.15 | 5 | 61.31 | −1 |
| Talavera de la Reina | 96 | 40.60 | 89 | 46.75 | 7 |
| Tarragona | 48 | 48.38 | 50 | 53.14 | −2 |
| Terrasa | 30 | 50.79 | 20 | 58.53 | 10 |
| Telde | 91 | 42.39 | 92 | 45.43 | −1 |
| Teruel | 86 | 43.06 | 94 | 45.13 | −8 |
| Toledo | 84 | 43.49 | 69 | 50.68 | 15 |
| Torrejón de Ardoz | 74 | 44.69 | 60 | 51.78 | 14 |
| Torrent | 45 | 48.92 | 36 | 55.67 | 9 |
| Torrevieja | 103 | 35.50 | 103 | 36.60 | 0 |
| Valencia | 34 | 50.10 | 53 | 52.85 | −19 |
| Valladolid | 19 | 51.75 | 24 | 57.86 | −5 |
| Vélez-Málaga | 99 | 39.82 | 97 | 42.74 | 2 |
| Vigo | 33 | 50.27 | 42 | 54.73 | −9 |
| Vitoria-Gasteiz | 1 | 61.05 | 1 | 68.33 | 0 |
| Zamora | 73 | 44.82 | 72 | 49.61 | 1 |
| Zaragoza | 2 | 58.72 | 2 | 66.60 | 0 |

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
