# Peer review of "Assessment of the Results and Methodology of the Sustainable Development Index for Spanish Cities"

_sustainability, doi:10.3390/su13116487_

Round 1

Reviewer 1 Report

The clarity of the article is jeopardized by lack of any description of the Sustainable Development Goals and related indicators. The extensive use of the acronyms SDG 1, SDG 2, etc. hinders the overall understanding of the study.

Section 2 does not adequately describe the methodology developed in the research.

Section 3 refers to indicators that are not described (listed in Spanish in the appendix) and is reduced to mere statistical argumentation, without allowing the reader to actually understand which indicators the authors are referring to. 

Overall, I found the paper to be quite interesting. Nonetheless, it presents an apparently extremely abstract study, without any concrete reference to the territory's characteristics.
The introduction of concrete examples, referred to the area under investigation, is suggested. 

Reviewer 2 Report

Congratulation for your work! In the following, I suggest some improvements.

The title should be better reflected by the content. I expect as a final result of your study:

  • SDI results' assessment - - with critical aspects and proposal for improvements (per cities, urban areas of ES??!!)
  • SDI methodology assessment - with critical aspects and proposal for improvements

Criticism of SDI results (presentation and use!!)/methodology could improve the paper contribution to scholarship.

Abstract should be re-written ... please do not use abbreviations in this part of the paper because it is difficult to follow the ideas.

In chapter 1, I suggest to extend your argumentation with the "urban sustainability" topic. It seems that you refers to this (see line 133 where you mention system of urban indicators", lines 473-474). Also, attention should be given to "sustainability awareness" as an implication of SDI assessment.

Regarding the paper/research objective (line 135-137) ... I suggest to carefully consider this with respect to the paper title.

Chapter 2 should be extended. There are missing the explanations of the tools and the statistical indicator used (detailed description of the theoretical approach). Why you have used these and not other? Please provide similar approaches in the literature. Lines 150-163 are not motivated! Why you do so? Lines 157 and 161 refers to "assumptions" ... and there is no research stage referring to assessment of SDI?

In Fig 1 ... values/numbers are incorrectly written (you should use point not coma). I suggest transforming Fig 1 into a correct data table. Similar to Fig 2, Table 1 - 14. I suggest reducing the no. of Tables and Figures because the scientific discourse is difficult to be followed. More Appendixes should be defined.

In my opinion, the statistical data processing should be carefully re-checked and better explained in the paper. Also, I suggest leaving in the paper text only those Tables and Figures that are aligned with the research objectives. 

Appendix A consists of the description of the indicators in Spanish language ... you should add a column with the translation in English.

Chapter 4 should be entitled "Final Conclusions and Discussions". It should be extended with comments regarding the way in which the research objectives have been achieved (better explained the assessment process of SDI by summarizing the main findings).

The conclusions must be better supported by other results presented in the literature.

It should be described the implications of the research results not only for the research perspective (increasing urban sustainability through monitoring developed by SDI), but also for policy-makers, public institutions, public services providers, citizens and other stakeholders. 
